# Near-infrared dual bioluminescence imaging in mouse models of cancer using infraluciferin

Cassandra L Stowe[1,2], Thomas A Burley[3], Helen Allan[4], Maria Vinci[3], Gabriela Kramer-Marek[3], Daniela M Ciobota[3], Gary N Parkinson[5], Tara L Southworth[6], Giulia Agliardi[1], Alastair Hotblack[1], Mark F Lythgoe[2], Bruce R Branchini[6], Tammy L Kalber[2], James C Anderson[4]*, Martin A Pule[1]*

[1]Cancer Institute, University College London, London, United Kingdom; [2]Centre for Advanced Biomedical Imaging, University College London, London, United Kingdom; [3]The Institute of Cancer Research, London, United Kingdom; [4]Department of Chemistry, University College London, London, United Kingdom; [5]School of Pharmacy, University College London, London, United Kingdom; [6]Department of Chemistry, Connecticut College, New London, United States

*For correspondence:
j.c.anderson@ucl.ac.uk (JCA);
m.pule@ucl.ac.uk (MAP)

**Competing interests:** The authors declare that no competing interests exist.

**Abstract** Bioluminescence imaging (BLI) is ubiquitous in scientific research for the sensitive tracking of biological processes in small animal models. However, due to the attenuation of visible light by tissue, and the limited set of near-infrared bioluminescent enzymes, BLI is largely restricted to monitoring single processes in vivo. Here we show, that by combining stabilised colour mutants of firefly luciferase (FLuc) with the luciferin ($LH_2$) analogue infraluciferin ($iLH_2$), near-infrared dual BLI can be achieved in vivo. The X-ray crystal structure of FLuc with a high-energy intermediate analogue, 5'-O-[N-(dehydroinfraluciferyl)sulfamoyl] adenosine (iDLSA) provides insight into the FLuc-$iLH_2$ reaction leading to near-infrared light emission. The spectral characterisation and unmixing validation studies reported here established that $iLH_2$ is superior to $LH_2$ for the spectral unmixing of bioluminescent signals in vivo; which led to this novel near-infrared dual BLI system being applied to monitor both tumour burden and CAR T cell therapy within a systemically induced mouse tumour model.

DOI: https://doi.org/10.7554/eLife.45801.001

## Introduction

Bioluminescence imaging (BLI) is used extensively for the sensitive, longitudinal and high-throughput monitoring of biological processes in vivo (*Xu et al., 2016*; *Paley and Prescher, 2014*; *Mezzanotte et al., 2017*; *Yao et al., 2018*; *Yeh and Ai, 2019*). Bioluminescence light emission is produced through the catalysis of a small molecule substrate, most commonly D-luciferin (D-$LH_2$), by a luciferase enzyme. The mutagenesis of bioluminescent enzymes has improved the sensitivity and accuracy of BLI in small animals (*Branchini et al., 2010*), (*Iwano et al., 2018*). However, despite its widespread use in scientific research, BLI is still largely restricted to tracking a single parameter in vivo. The ability to discretely monitor two biological parameters (dual-BLI) simultaneously within a single animal is highly desirable (*Xu et al., 2016*) with potential uses ranging from the monitoring of tumour burden alongside cellular therapy, to the visualisation of dynamic biological processes such as protein-protein interactions (*Prescher and Contag, 2010*).

Previous approaches to dual-BLI have been disappointing. The use of multiple bioluminescent proteins which catalyse different substrates is the most frequently used method but suffers from a

**eLife digest** Fireflies and some other insects glow to attract mates or prey. This so-called bioluminescence occurs when an enzyme called luciferase modifies the molecule luciferin, which can then emit bright yellow-green light. The gene that encodes the luciferase enzyme has been introduced into cells from mammals, including cancer cells. In the presence of luciferin, these cells begin to glow. The brightness of the bioluminescence depends on how many cancer cells are growing and dividing. The light is bright enough for the cancer cells making luciferase to be transplanted into mice so their behaviour can be examined.

However, blood and other tissues absorb the yellow-green light, making it hard to see the cancer cells deep within a mouse. To circumvent this problem, researchers designed a new type of luciferin, called infraluciferin, which emits red light that shines through blood and tissues. There are now also different variants of the luciferase enzyme, which act on infraluciferin to make different shades of red light.

Stowe et al. wanted to test if two different biological events happening at the same time could be observed using two shades of bioluminescent red in a single live mouse. First, a mixture of cancer cells containing two versions of luciferase were transplanted into mice. When the mice were then given infraluciferin, the two types of cancer cells could be distinguished based on the different shades of red bioluminescence. In a second experiment, Stowe et al. tracked the treatment of cancer cells with immune cells, by introducing a different version of luciferase into each of the two groups of cells. Over time, the red light produced by the immune cells grew stronger than that of the cancer cells, indicating that the number of cancer cells had decreased and that the treatment was effective.

Together, this work shows that it can be simple, cheap and efficient to observe more than one cell type, or even disease, in a living system. This technique may be used by scientists to study different diseases and treatment options in mice. Importantly, it will also reduce the number of animals used to do this research.

DOI: https://doi.org/10.7554/eLife.45801.002

number of limitations. Sequential substrate administration is normally required, in addition this method commonly employs a combination of a coelenterazine and a $D-LH_2$ utilising luciferase (with the blue emission from the former being heavily absorbed compared to the yellow-green emission from the latter) (*Maguire et al., 2013*), (*Stacer et al., 2013*). Differences in biodistribution and reaction kinetics of two substrates can make image co-registration and interpretation difficult. The development of orthogonal luciferase-luciferin pairs has solved some of these limitations but this approach still requires multiple substrate administrations (*Rathbun et al., 2017*).

An ideal dual-BLI approach would use two spectrally distinct bioluminescent proteins utilising a single substrate followed by spectral unmixing of the signal. However, this approach is not currently feasible using $LH_2$. Although luciferases can be mutated to alter the colour of their emission, a limit appears to have been reached for mutational colour modulation of firefly and related luciferases. The most red-shifted of these enzymes have maximal peak emissions between 610 and 620 nm (*Branchini et al., 2010*). This is insufficient for dual BLI in vivo. Due to the differential attenuation of light by biological tissue spectral unmixing of a red-shifted luciferase paired with a standard or green-shifted enzyme is challenging, especially in deeper tissue models (*Mezzanotte et al., 2011*). Shifting the emission of both enzymes into the near infrared must be achieved to allow adequate unmixing. To further red-shift peak emission we and others have turned to chemical modification of the D-luciferin ($LH_2$) substrate (*Adams and Miller, 2014*). We recently described the $LH_2$ analogue infraluciferin ($iLH_2$) which has a luciferase dependent red-shifted peak emission of up to 706 nm (*Jathoul et al., 2014*) (*Figure 1a*). We hypothesised that combining colour modulation of bioluminescence through mutagenesis of the FLuc protein along with red-shifting bioluminescence by chemical modification of $LH_2$ would allow dual-BLI, an approach that has not been described previously.

In this work, we explored the possible structural interactions in the enzyme that may account for the near infrared emission of iLH2 and its application to dual-BLI in vivo. First, the X-ray crystal structure of FLuc in complex with a high-energy intermediate analogue, 5'-O-[(N-dehydroinfraluciferyl)-

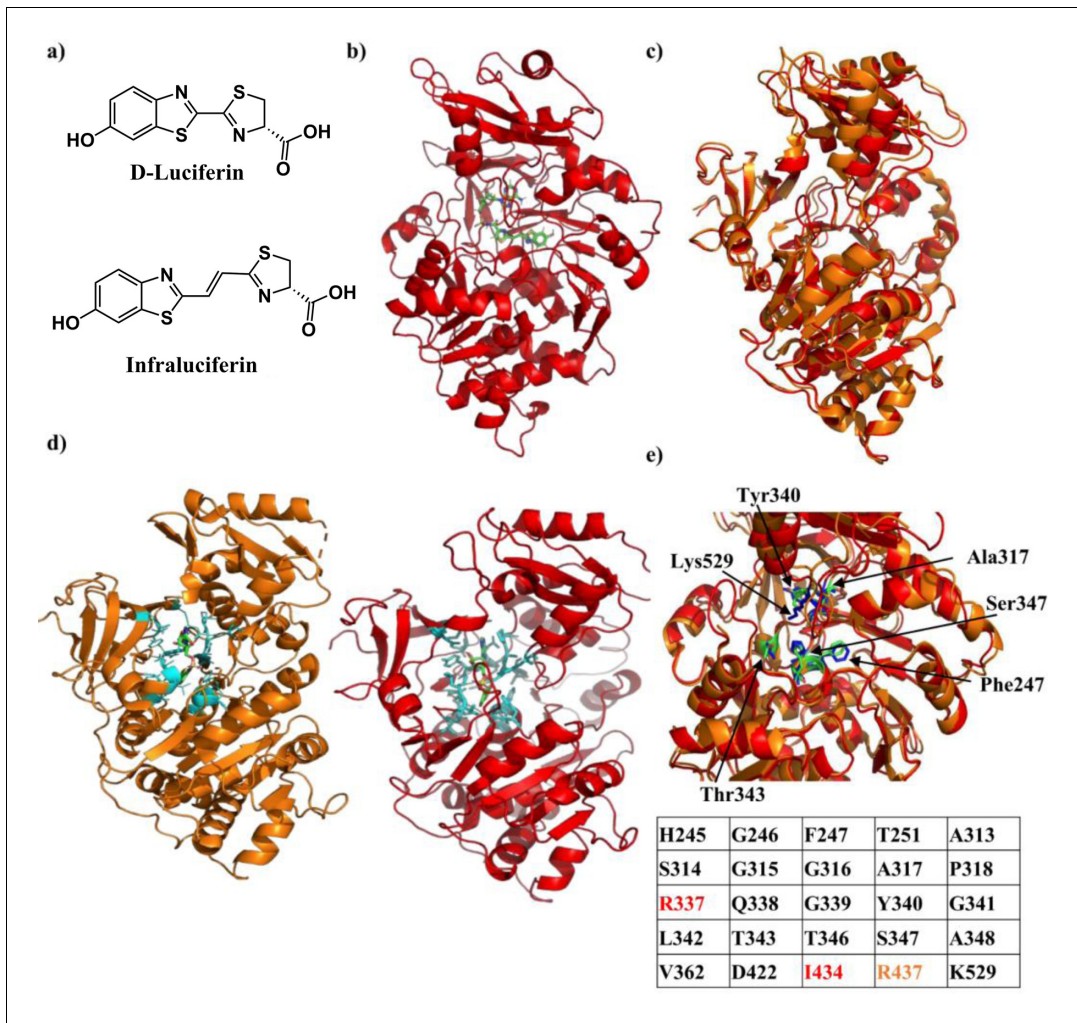

**Figure 1.** Crystal structure of Firefly luciferase in complex with a iLH$_2$ analogue. (a) Chemical structures of native D-Luciferin (LH$_2$) and the LH$_2$ analogue infraluciferin (iLH$_2$). (b) The crystal structure of Firefly luciferase (FLuc) in complex with the infraluciferyl-adenylate analogue 5'-O-[(N-dehydroinfraluciferyl)-sulfamoyl] adenosine (iDLSA) resolved to a 3.2 Å resolution (PDB ID: 6HPS). (c) The structure of FLuc in complex with the iLH$_2$ analogue iDLSA aligned to the reported structure of FLuc in complex with the LH$_2$ analogue 5'-O-[(N-dehydroluciferyl)-sulfamoyl] adenosine (DLSA) (PDB ID: 4G36) (*Sundlov et al., 2012*) based on the FLuc N-terminal domain (residues 1–436). The structure of FLuc in complex with iLH$_2$ is shown in red, and the structure of FLuc in complex with LH$_2$ is shown in orange. (d) The structure of FLuc in complex with luciferin analogue DLSA (PDB ID: 4G36) (orange) and the infra-luciferin analogue iDLSA (red). Those residues within 4 Å of the substrate in each structure are highlighted in blue. (e) The table lists all residues within 4 Å of the both substrates, with those in orange or red only being found within 4 Å of DLSA and iDLSA respectively. (f) Highlights the same seven active site residues for both the aligned structures, with FLuc iLH$_2$ residues in blue and FLuc LH$_2$ residues in green. All analysis performed in PyMOL software (Schrodinger).

DOI: https://doi.org/10.7554/eLife.45801.003

The following figure supplement is available for figure 1:

**Figure supplement 1.** Synthetic Scheme for the synthesis of iDLSA.

DOI: https://doi.org/10.7554/eLife.45801.004

sulfamoyl] adenosine (iDLSA) was determined to provide insight into the FLuc-iLH$_2$ light-emitting reaction. Next, we selected a pair of stabilised colour-shifted FLuc mutants, which emit with a 20 nm separation in peak emission wavelength with iLH$_2$ in the near infrared. We demonstrated the ability to spectrally unmix these two biological signals in vivo at depth using iLH$_2$. Finally, we show a proof-

of-concept of utility using this novel dual imaging technique to longitudinally monitor both tumour burden and chimeric antigen receptor (CAR) T cell therapy within a single animal model.

## Results

### Crystal structure of firefly luciferase in complex with a $iLH_2$ analogue

To help understand the red shift in bioluminescence emission from FLuc that is observed in its reaction with $iLH_2$, the X-ray crystal structure of FLuc in complex with iDLSA was resolved and is shown in *Figure 1b* (PDB ID: 6HPS). Data collection and refinement statistics (molecular replacement); and data collection, phasing and refinement statistics for mad (semet) structures can be found in *Figure 1—figure supplement 1*. iDLSA captures FLuc in the adenylation step of the light emitting reaction ($^1H$ and $^{13}C$ data spectra synthetic chemical compounds can be found in). The conformation of the $iLH_2$ heterocyclic rings with respect to the alkene, as drawn in *Figure 1a*, is confirmed to be as predicted by computational studies and is the most likely conformation of the light emitting form (*Berraud-Pache and Navizet, 2016*). This newly crystallised FLuc structure was aligned with the reported structure of FLuc with 5'-O-[(N-dehydroluciferyl)-sulfamoyl] adenosine (DLSA) (PDB ID: 4G36) (*Sundlov et al., 2012*). The structures show good alignment to each other, however there is evidence of a more open active site supported by a reduction in root-mean-squared (RMSD) score when aligned based on just the N-terminal domain of FLuc rather than the entire structure (RMSD = 0.688 and 0.783 respectively) (*Figure 1c*).

All FLuc residues in close proximity (4 Å) to DLSA were also found to be within the same distance to iDLSA, with the exception of Arg437, >4 Å away from iDLSA (*Figure 1d and e*). We noted that despite differences in the conformation of iDLSA compared to DLSA in both 4G36 and *L. cruciata* 2D1S (*Nakatsu et al., 2006*) structures, the positions of the phenolic groups are quite similar (~0.5 Å). The altered position of the benzothiazole ring and the greater size of iDLSA may be the cause of a series of small active site changes that affect residues Glu311, Arg337, Asn338, Gly339, and Thr343 resulting in a total of six differences in H-bonding interactions. When specific residues implicated in the light emitting reaction (*Sundlov et al., 2012*) were measured between the two structures differences ranged from 0.7 to 1.6 Å; with the biggest divergence being Lys529 (found in the C-terminal cap) which had a 2.4 Å difference in the nitrogen residue found in the side chain of the amino acid (*Figure 1f*). The resulting increase in active site polarity due to the rotation of the C-terminal cap, if maintained during the light emitting conformation, could contribute to the red-shift in light emission (*Nakatsu et al., 2006*), in addition to the increased π-conjugation through the chemical structure of the emitter. This X-ray structure will help the future design of more efficient FLuc-$iLH_2$ pairs.

### Spectral unmixing of firefly luciferase mutants in vitro

A range of colour-shifted, thermo- and pH stable FLuc mutants were spectrally characterised in vitro with a comparative selection of $LH_2$ analogues proven to red-shift bioluminescence emission (CycLuc1– *Evans et al., 2014*; Aka-Lumine-HCL – *Kuchimaru et al., 2016*; and $iLH_2$– *Jathoul et al., 2014*). Two new luciferins $NH_2$-$NpLH_2$ and OH- $NpLH_2$ have also been shown to have near infrared emissions (*Hall et al., 2018*) but these were reported too late to include in this study. FLuc mutants were engineered to combine mutations reported to provide superior stability (*Jathoul, 2012*) and colour-shifting capability (*Branchini et al., 2005*) (stabilising and colour shifting FLuc mutations are detailed in Materials and methods). The Raji B lymphoma cell line engineered to express a FLuc mutant were spectrally imaged after addition of each substrate. These cell lines were subsequently used for all in vitro and in vivo testing. Both CycLuc1 and Aka-Lumine-HCL showed a consistent red-shift in peak bioluminescence emission wavelength to ~600 nm and ~660 nm respectively for all FLuc mutants, making these substrates unsuitable for dual colour BLI (*Figure 2—figure supplement 1*). The data confirmed that with $LH_2$ both FLuc_natural and FLuc_green have a peak emission of ~560 nm, whilst FLuc_red has a peak emission of ~620 nm (*Figure 2—figure supplement 1*) (*Jathoul, 2012*), (*Branchini et al., 2005*). When tested with $iLH_2$ all FLuc mutants were shifted >100 nm into the near infrared but maintained their relative spectral shift [FLuc_green ~ 680 nm, FLuc_natural ~ 700 nm and FLuc_red ~ 720 nm (*Figure 2—figure supplement 1*)]. From this, we

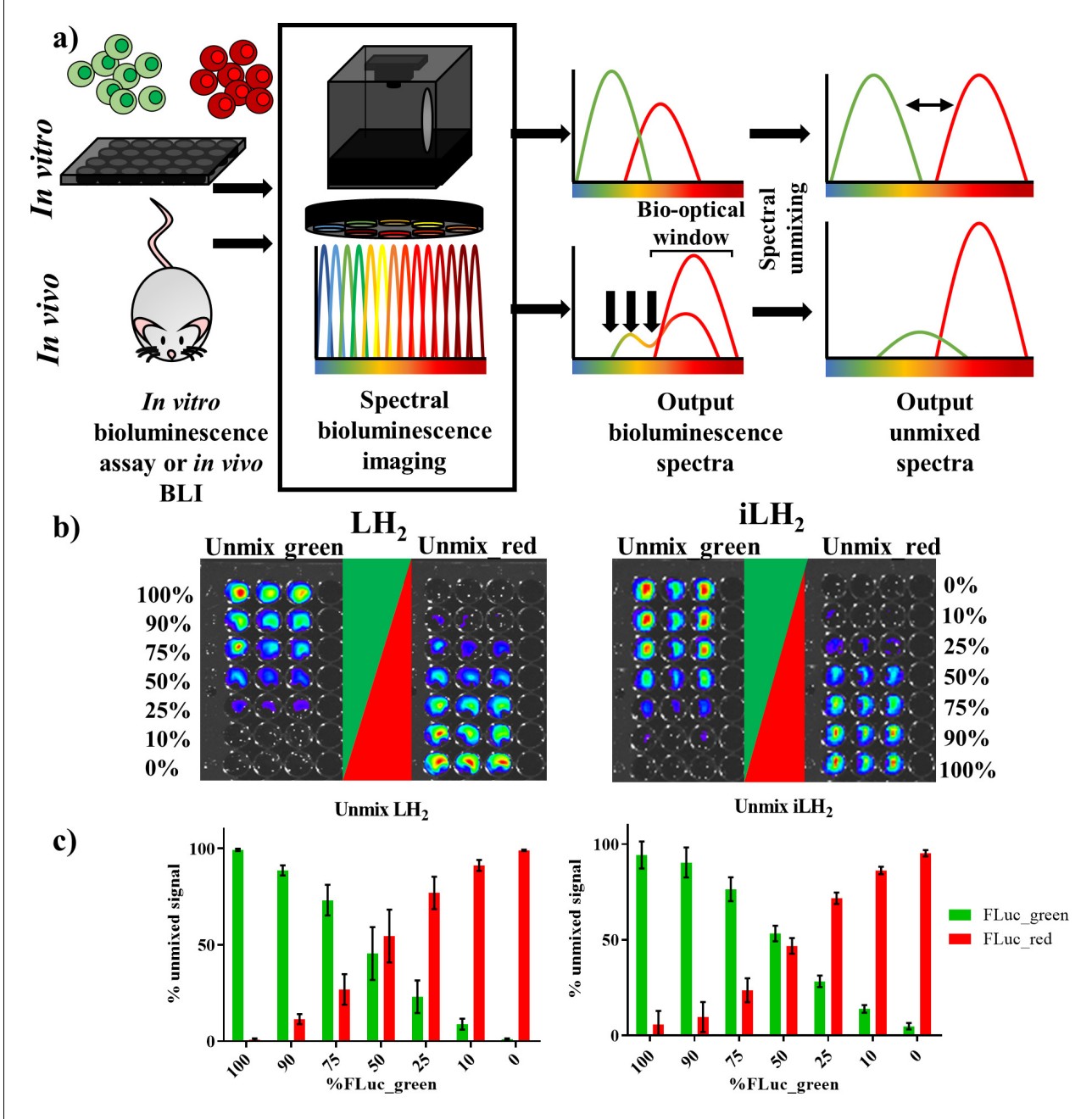

**Figure 2.** Spectral unmixing of Firefly luciferase mutants in vitro. (a) Bioluminescence spectral unmixing of cells expressing colour-shifted Firefly luciferase (FLuc) mutants, for both in vitro assays and in vivo animal models, requires spectral bioluminescence imaging through a series of bandpass filters. Bioluminescence spectral imaging acquires spectral data, which can then be deconvoluted into its separate components. As depicted, spectral unmixing in vivo is significantly more challenging due to attenuation of bioluminescent signal that does not fall within the bio-optical window. (b) FLuc colour mutants FLuc_green and FLuc_red expressed in the B lymphoma Raji cell line were mixed in various proportions (0–100% of the total population). After addition of either D-luciferin (LH$_2$) or infraluciferin (iLH$_2$) plates were spectrally imaged using the IVIS Spectrum (Perkin Elmer). Unmixed green and unmixed red output images produced from library spectral unmixing using Living Image software (Perkin Elmer) for both substrates. (c) Percentage unmixed signal of FLuc_green and FLuc_red for each ratio of FLuc expressing cells tested when imaged with LH$_2$ and iLH$_2$. Signal adjusted to 100% populations. Mean and standard deviation plotted (n = 6 for both LH$_2$ and iLH$_2$).

DOI: https://doi.org/10.7554/eLife.45801.005

The following figure supplements are available for figure 2:

**Figure supplement 1.** Bioluminescence spectra of x11 FLuc mutants FLuc_natural, FLuc_green and FLuc_red with LH$_2$ analogues.

DOI: https://doi.org/10.7554/eLife.45801.006

*Figure 2 continued on next page*

*Figure 2 continued*

**Figure supplement 2.** Spectral unmixing of different levels of expressed Fluc_green and Fluc_red in cells with $LH_2$ and $iLH_2$.

DOI: https://doi.org/10.7554/eLife.45801.007

progressed further with two FLuc mutants, FLuc_green and FLuc_red to explore their utility for dual-BLI.

The ability to spectrally unmix FLuc_green and FLuc_red (*Figure 2a*) in vitro was investigated by mixing the two FLuc_mutants expressed in the Raji B lymphoma cell line at various ratios followed by spectral imaging and unmixing with both $LH_2$ and $iLH_2$ (*Figure 2b*). As would be expected from accurate spectral unmixing, the top wells were classified as containing mostly FLuc_green signal, which gradually decreased down the plate in line with the decreasing proportions of FLuc_green expressing cells, with the bottom wells being largely classified as FLuc_red signal for both $LH_2$ and $iLH_2$. The percentage unmixed signal of FLuc_green and FLuc_red was plotted for each ratio of FLuc expressing cells (*Figure 2c*). Correlation analysis was performed on this data comparing input cellular proportions with unmixed signal, giving $R^2$ values of 0.9983 and 0.9972 for $LH_2$ and $iLH_2$ respectively. Even though all 18 bandpass filters equipped on the IVIS Spectrum were utilised for spectral unmixing in this in vitro testing, we appreciate that not all potential users of this novel dual bioluminescence methodology will have access to machines with such a wide selection of bandpass filters. Therefore, further analysis of our data showed that spectral unmixing could be achieved with high accuracy just using a subset of filters. For $LH_2$, the use of 3 bandpass filters (500 nm, 660 nm, 820 nm) gave an $R^2$ value of 0.9958; For $iLH_2$, the use of 3 bandpass filters (600 nm, 700 nm, 800 nm) gave an $R^2$ value of 0.9937. The highest accuracy of spectral unmixing we could achieve using just two filters were $R^2$ values of 0.9776 and 0.9775 for $LH_2$ (500 nm and 720 nm) and $iLH_2$ (600 nm and 720 nm), respectively. Additionally, an experiment was carried out where FLuc_green and FLuc_red have been expressed at different levels in the same cell. Spectral bioluminescence imaging and unmixing has subsequently been performed to successfully reflect these differing expression levels with both $LH_2$ and $iLH_2$ (*Figure 2—figure supplement 2*). This spectral imaging data show that both $LH_2$ and $iLH_2$ can be used for dual bioluminescence reporting in vitro.

## Spectral characterisation of firefly luciferase mutants with $LH_2$ and $iLH_2$ in vivo

To investigate the use of FLuc_green and FLuc_red with $LH_2$ and $iLH_2$ for in vivo dual BLI three NOD *scid* gamma (NSG) tumour models, representing increasing tissue depth (subcutaneous, systemic and intracranial), were established with the Raji B lymphoma cell line expressing either FLuc_green or FLuc_red (as described for in vitro experiments). All tumour models were then spectrally imaged with both $LH_2$ and $iLH_2$ (*Figure 3*, *Figure 3—figure supplement 1* and *Figure 3—figure supplement 2*) to obtain the normalised spectra and average radiance of each FLuc mutant with both luciferins in all three in vivo models (*Figure 4*). The normalised bioluminescence spectra for every mouse in each model when imaged with $LH_2$ is shown, with the total radiance for each mouse plotted to the right of the spectral plot (*Figure 4a–c*). The data show that when imaged with $LH_2$ both FLuc_green and FLuc_red had an average peak emission between 610–630 nm; meaning the peak emission for FLuc_red is maintained as in vitro whereas the peak emission of FLuc_green is red shifted by ~60 nm in vivo. FLuc_green also exhibited a bimodal spectral distortion, with a minor peak at ~560 nm. In contrast to $LH_2$, FLuc_green and FLuc_red had a ~ 20 nm separation of average peak emissions in all three animal models with $iLH_2$ (FLuc_green ~ 700 nm and FLuc_red ~720 nm) (*Figure 4d–f*).

In addition to the separation of peak emission wavelengths when imaged with $iLH_2$ in vivo, the relative intensities of FLuc_green and FLuc_red were more comparable when imaged with $iLH_2$ than with $LH_2$; With $LH_2$ FLuc_red had an average radiance that was 42 (subcutaneous), 4.12 (systemic) and 7.28 (intracranial) times brighter than FLuc_green (*Figure 4a–c*). Whereas, when imaged with $iLH_2$ the average radiance between FLuc_green and FLuc_red was 1.38 (subcutaneous), 1.2 (systemic) and 1.51 (intracranial) times different (*Figure 4d–f*). No statistically significant difference in relative intensities between FLuc_green and FLuc_red was found in tumour models imaged with $iLH_2$ (p=0.3414, 0.4594 and 0.6153 for the subcutaneous, systemic and intracranial tumour models

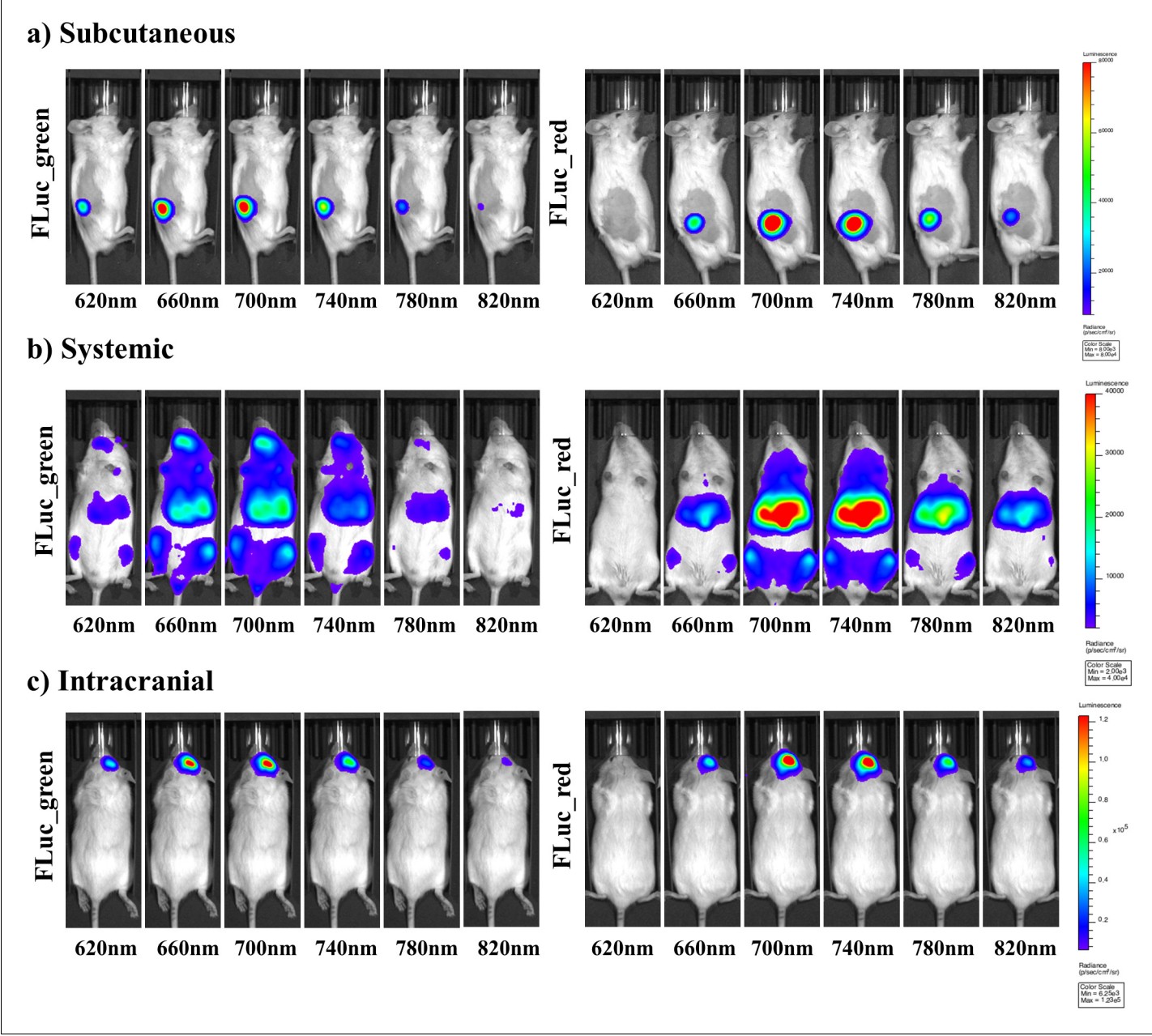

**Figure 3.** Representative selection of bioluminescent images of FLuc mutants with iLH$_2$ in vivo. A representative selection of filter images from mice engrafted with the Raji B lymphoma cell line expressing either Fluc_green or Fluc_red for each of the in vivo tumour models when imaged with iLH$_2$ (**a**) subcutaneous (**b**) systemic and (**c**) intracranial (bandpass filters not shown are 600 nm, 640 nm, 680 nm, 720 nm, 760 nm, 800 nm).
DOI: https://doi.org/10.7554/eLife.45801.008

The following figure supplements are available for figure 3:

**Figure supplement 1.** Example of all filter images for FLuc_green and FLuc_red acquired in each in vivo model with iLH$_2$.
DOI: https://doi.org/10.7554/eLife.45801.009
**Figure supplement 2.** Corresponding example of all filter images for FLuc_green and FLuc_red acquired in each in vivo model with LH$_2$.
DOI: https://doi.org/10.7554/eLife.45801.010

respectively, T test). This comparability of relative intensities between FLuc_green and FLuc_red with iLH$_2$ means that if used as genetic reporters for dual imaging, the dynamic range of radiance values for both enzymes will be more similar, therefore giving a more accurate comparison of the processes being monitored.

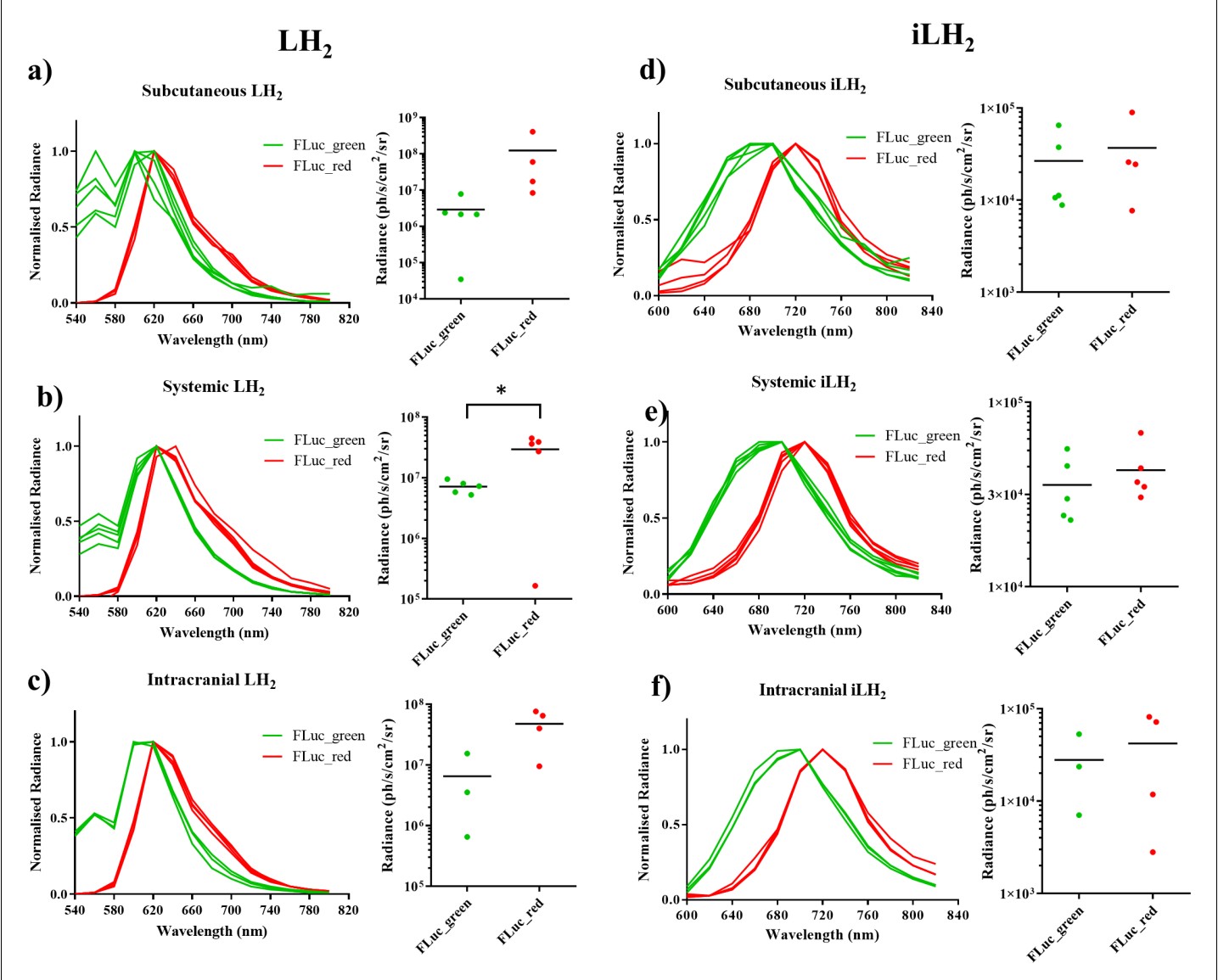

**Figure 4.** Spectral characterisation of FLuc mutants with LH₂ and iLH₂ in vivo. The normalised spectra, and a plot of average radiance, for each FLuc mutant when expressed in the Raji B lymphoma cell line engrafted in each tumour model when imaged with LH₂ is shown, (**a**) subcutaneous (**b**) systemic and (**c**) intracranial, and when the same animals were imaged with iLH₂ is shown, (**d**) subcutaneous (**e**) systemic and (**f**) intracranial. Subcutaneous (n = 9), systemic (n = 10), intracranial (n = 7). T test used to establish statistical significance comparing average radiance values (systemic model when imaged with LH₂ p=0.0224).

DOI: https://doi.org/10.7554/eLife.45801.011

## Spectral unmixing of firefly luciferase mutants in vivo

To validate the ability to spectrally unmix FLuc_green and FLuc_red in vivo with iLH₂ a systemic Raji tumour model was established. Raji cell lines expressing the FLuc mutants were mixed in the following ratios: 90:10, 75:25 and 50:50 for FLuc_green: FLuc_red and vice versa. After spectral BLI with both substrates, animals were sacrificed and the bone marrow was extracted for flow cytometry analysis to confirm the proportions of engrafted Raji FLuc populations (representative examples of flow cytometry plots can be found in *Figure 5—figure supplement 1*). Spectral unmixing was performed using Living Image (*Perkin Elmer*) by creating library spectra of FLuc_green and FLuc_red with both

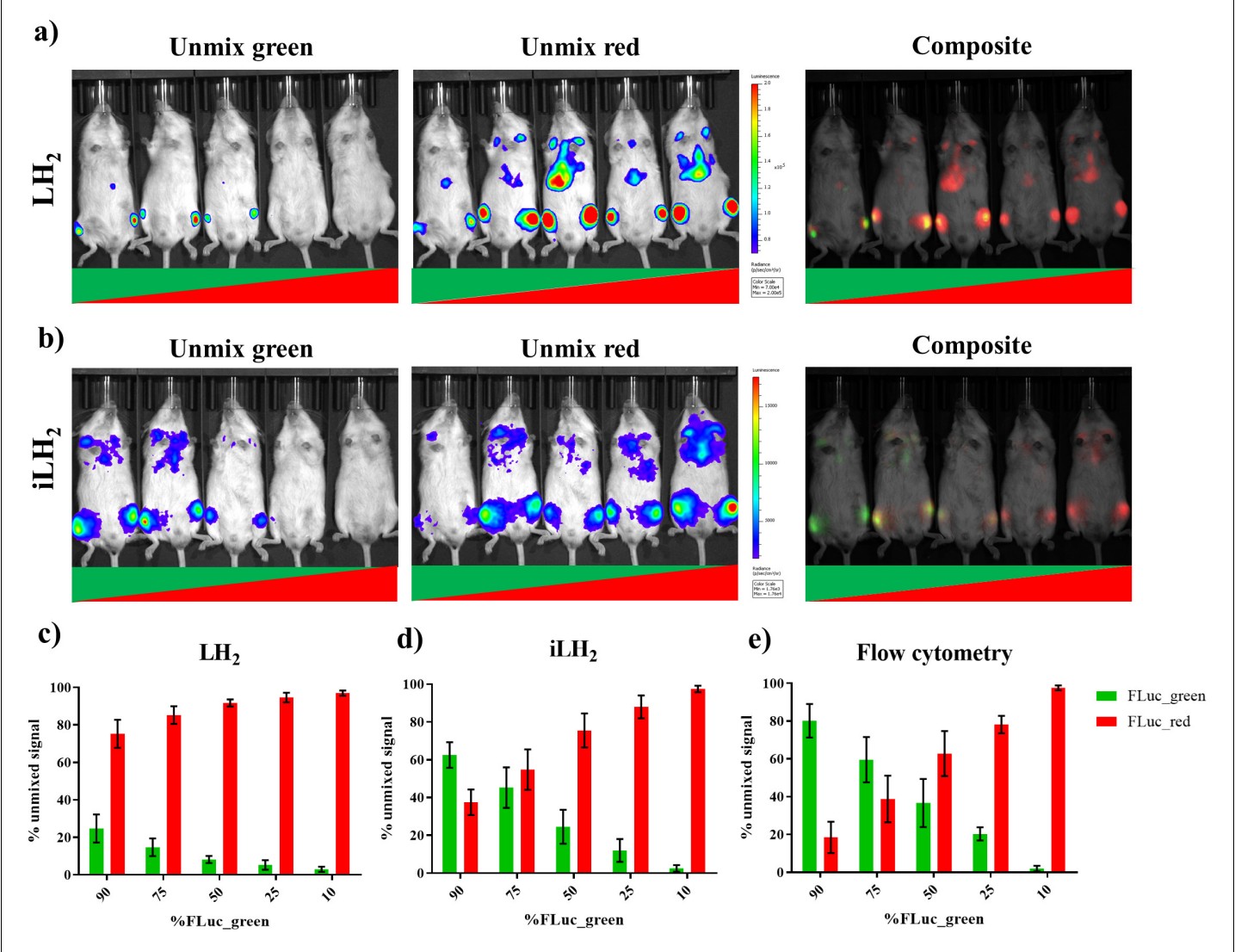

**Figure 5.** Spectral unmixing of Firefly luciferase mutants in vivo. The Raji B lymphoma cell line expressing either FLuc_green or Fluc_red, were mixed in various proportions and engrafted in a systemic in vivo model. Cell mixtures ranged from 90:10 to 10:90 Fluc_green to Fluc_red, and each imaging session included one mouse engrafted with each mixture. Animals were spectrally imaged with D-Luciferin (LH$_2$) and infraluciferin (iLH$_2$) using the IVIS spectrum (Perkin Elmer) in separate imaging sessions. (a) An example of the unmixed Fluc_green, umixed Fluc_red and composite output images when imaged with LH$_2$ and (b) iLH$_2$. Percentage of unmixed Fluc_green and Fluc_red signal for each cell mixture when imaged with (c) LH$_2$ (d) iLH$_2$, and (e) extracted bone marrow samples when analysed by flow cytometry (n = 4 per dilution condition).

DOI: https://doi.org/10.7554/eLife.45801.012

The following figure supplement is available for figure 5:

**Figure supplement 1.** Representative flow cytometry plots of extracted bone marrow samples.

DOI: https://doi.org/10.7554/eLife.45801.013

LH$_2$ and iLH$_2$, established from the pure expressing populations obtained during in vivo spectral characterisation (*Figure 4*). Output images with both LH$_2$ (*Figure 5a*) and iLH$_2$ (*Figure 5b*) were generated for FLuc_green and FLuc_red, as well as a composite image for each substrate.

The percentage signal unmixed as FLuc_green and FLuc_red with both LH$_2$ (*Figure 5c*) and iLH$_2$ (*Figure 5d*) was determined and correlated to the percentage population of each FLuc_mutant within the Raji cell population taken from extracted bone marrow samples and analysed using flow

cytometry (*Figure 5e*). A correlation of 0.99 was found with $iLH_2$ ($R^2$ value, SD = 0.01). $LH_2$ had a correlation of 0.89 ($R^2$ value, SD = 0.06), which was significantly different from the $R^2$ values obtained by flow cytometry (p<0.0001, ONE-Way ANOVA with post hoc Tukey test). No significant difference was found between $R^2$ values determined by flow cytometry and unmixed bioluminescence signal using $iLH_2$. Additionally, significant differences were found between the percentage unmixed signal using $LH_2$ and cellular proportions determined by flow cytometry, with p values of < 0.0001, 0.003, 0.0042 and 0.0056 for the 90%, 75%, 50% and 25% FLuc_green conditions respectively. No significant difference was found between percentage unmixed signal using $iLH_2$ and cellular proportions determined by flow cytometry, except for the 90% FLuc_green condition (p=0.0257). Therefore, $iLH_2$ is superior to $LH_2$ for in vivo dual reporting applications.

## Application of dual bioluminescence imaging using infraluciferin

The characterised and validated in vivo dual BLI system using FLuc mutants in combination with $iLH_2$ was then applied to track tumour burden and CAR T cell therapy within the same animal model. A

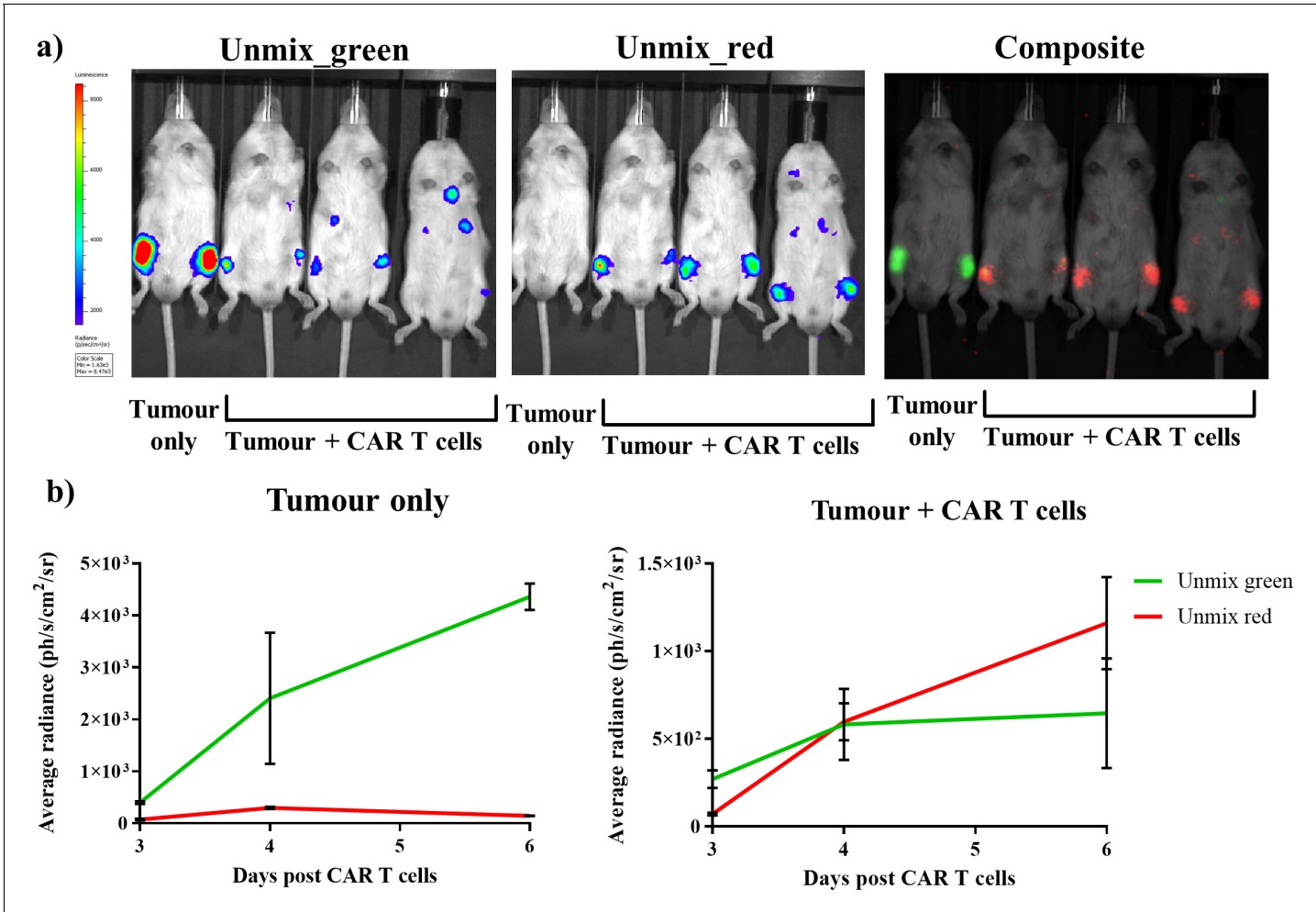

**Figure 6.** Dual bioluminescence imaging of CAR T cell therapy using infraluciferin. Mice were engrafted with the Raji B lymphoma tumour cell line expressing Fluc_green, and were subsequently treated with healthy human donor T cells engineered to express CD19 CAR and FLuc_red (except the tumour only control). Animals were then spectrally imaged after administration of iLH2 using the IVIS spectrum (Perkin Elmer). (a) The unmixed Fluc_green images, representing tumour burden, and unmixed Fluc_red images, representing CAR T cell homing, and the composite image are shown for day six post CAR T cell treatment. (b) The average radiance of signal classified as Fluc_green and Fluc_red is plotted for days 3, 4 and 6 post CAR T cell administration for the tumour only control and treatment (tumour + CAR T cells) animals. Three mice were randomly selected to receive CAR T cell therapy after engraftment was confirmed. Mean and standard deviation plotted. Radiance values from each femur are treated separately (tumour only = 2, treatment = 6).

DOI: https://doi.org/10.7554/eLife.45801.014

Raji B lymphoma tumour cell line expressing FLuc_green was used, and healthy human donor T cells were engineered to express CD19 CAR and FLuc_red linked via a 2A peptide. Tumour cells were first systemically engrafted, followed by administration of CAR T cells 8 days later. A control animal received tumour only. Spectral BLI using iLH$_2$ was then performed at 3, 4 and 6 days post CAR T cell administration on the IVIS Spectrum.

Output images for the unmixed FLuc_green and FLuc_red signal, as well as the composite of the two unmixed images, for day six post CAR T cell administration were generated (*Figure 6a*). The radiance values from the unmixed FLuc_green and FLuc_red images for both tumour control and treatment (tumour + CAR T cells) at 3, 4 and 6 days post CAR T cell treatment were determined (*Figure 6b*). As would be expected, the tumour only control showed consistently high levels of FLuc_green signal which increased overtime compared to FLuc_red, which is likely due to the close proximity of Fluc_red photons emitted by the neighbouring treatment mouse. However, the CAR T cell treatment mice initially had a higher proportion of FLuc_green signal, which was then surpassed by FLuc_red by the final imaging. This represents the homing of CAR T cells expressing FLuc_red to the FLuc_green expressing Raji B lymphoma tumour, followed by the expansion of the CAR T cells and the reduction in the tumour growth in the treatment mice.

## Discussion

Small animal dual (or multi-parameter) bioluminescence is highly desirable. Currently, most dual BLI has been achieved with two luciferases each utilising a different substrate, one of which is normally the auto luminescent coelenterazine (*Maguire et al., 2013*). A simpler approach would be to use a single LH$_2$ substrate and two firefly (or related) luciferases which emit at different wavelengths. This in theory has advantages of higher quantum yield and a more favourable substrate. Such an approach has been attempted: for instance Mezzanotte et al tested dual BLI in vivo using LH$_2$ with the green luciferase (CBG99) with the red luciferase (Ppy RE8) (*Mezzanotte et al., 2011*). However, in deeper tissues the shorter wavelength component of green emitting enzyme emission is heavily attenuated by mammalian tissues leaving an 'in vivo spectrum' which is almost indistinguishable from that of the red luciferase.

To illustrate this, and as a control for subsequent experiments, we attempted dual BLI using a pair of stabilised firefly luciferases, which are green/red shifted to 546 nm and 610 nm respectively using LH$_2$ as a substrate (*Branchini et al., 2007*). Whilst in vitro the spectra could be easily distinguished (*Figure 2*) in vivo the spectral separation between the two FLuc enzymes was lost due to the differential attenuation of the green enzyme by biological tissue (*Figure 4*). An obvious solution is to red-shift both enzymes into the optic window while maintaining an adequate separation. However, although bioluminescence emission has been successfully red-shifted through mutagenesis of firefly luciferase the structure of the LH$_2$ substrate ultimately limits this approach. Dual BLI with a single substrate in the near-infrared should be better; this has been described using BRET based reporters with an coelenterazine derived substrate (*Rumyantsev et al., 2016*). To move bioluminescence into the near infrared modification of the LH$_2$ substrate was required. Indeed, red-shifted luciferin analogues have been reported. These include CycLuc1 and Aka-Lumine, however neither permit variation in emission spectra with different Luciferase mutants (*Evans et al., 2014*), (*Kuchimaru et al., 2016*; *Iwano et al., 2018*) (*Figure 2*). More recently, two Naphthyl luciferins, NH$_2$-NpLH$_2$ and OH-NpLH$_2$, were reported to red shift the in vitro bioluminescence emission of CBR to 664 nm and 758 nm respectively (emission max 614 nm with LH$_2$). Interestingly, NH$_2$-NpLH$_2$ was shown to have an in vitro peak emission of 730 nm with an optimised version of CBR (CBR2). The potential application of CBR and CBR2 with NH$_2$-NpLH$_2$ for dual-BLI in vivo was not explored; and the reported broad emission spectrum of optCBR2 in live cells is likely to make any near-infrared dual BLI with NH$_2$-NpLH$_2$ challenging, however this approach cannot be discounted (*Hall et al., 2018*).

We described previously iLH$_2$, which has a near-infrared emission and in contrast to other LH$_2$ analogues maintains the 6' hydroxyl benzothiazole group of LH$_2$ that preserves the colour-shifts of mutant luciferases (*Jathoul et al., 2014*). Analysis of the x-ray crystal structure of FLuc in complex with the iLH$_2$ analogue (iDLSA) revealed that the hydrogen-bonding network, thought to be critical in stabilising the phenolate ion of the emitter, is disrupted in the FLuc-iDLSA due to the accommodation of the larger iLH$_2$ substrate (*Branchini et al., 2017*). This enables full charge delocalization of

the phenolate ion, in addition to extended conjugation, resulting in a red-shifted emission as suggested by homology modelling experiments with the recently described near infrared emitting naphthyl analogue (*Hall et al., 2018*). Evidence for the disruption of the network is the absence of a H-bond between Arg337 and Glu311 that is found in the DLSA structures of FLuc and *L. cruciata* Luc. This H-bonding interaction has been proposed to be important for stabilising an active site conformation for green light emission (*Viviani et al., 2005*). In addition, the resulting increase in active site polarity due to the rotation of the C-terminal cap could also contribute to the red-shift in light emission (*Nakatsu et al., 2006*). It must be noted that the crystal structure reported here has captured FLuc in the adenylation step of the reaction, therefore further computational modelling/crystal structure elucidation would be expected to provide further information on the light-emitting step of the reaction with $iLH_2$ (*Nakatsu et al., 2006*). Finally, the more open adenylation conformation of FLuc -iDLSA may affect the adenylate significantly, altering light production by decreasing the yield of the electronically excited state emitter and/or the efficiency in which the emitter produces a photon. Moreover, small differences in the binding position of the adenylate seen here, caused by positional changes of key active site residues could have a similar effect.

Given that $iLH_2$ preserves the colour modulation of mutant luciferases (in addition to the 100 nm red shift), we set about to explore dual-BLI with $iLH_2$. We identified two stabilised FLuc enzymes with colour shifting mutations: FLuc_green (V241I/G246A/F250S) and Fluc_red (S284T). Both exhibit a wide separation in peak emission wavelength, and were also balanced in their relative intensities. We next compared in vitro spectra with in vivo spectra from luciferase expressing cells implanted subcutaneously, systemically and intracranially (to approximate superficial, intermediate and deep light source) with both $LH_2$ and $iLH_2$. The near-infrared emission of the FLuc mutants with $iLH_2$ meant spectral separation was maintained for all animal tumour models. Importantly, the spectra of the two FLuc mutants remained consistent over tissue depth, meaning this dual BLI system could be applied to animal models without prior spectral characterisation. This maintenance of spectral separation, and similarity of relative intensities, between FLuc_green and FLuc_red when imaged with $iLH_2$ meant this near-infrared bioluminescence system was found to be significantly better at spectral unmixing in vivo than using $LH_2$. This was demonstrated when the unmixed bioluminescent signals were correlated with actual cellular populations, determined by flow cytometry, ($R^2 = 0.99$ and 0.89 for $iLH_2$ and $LH_2$ respectively).

One potential limitation of this system in its current state is the lower quantum yield of the FLuc-$iLH_2$ reaction when compared to the FLuc-$LH_2$ reaction,~2–3 orders of magnitude dimmer. However, in this study, all mice were successfully spectrally imaged with both $LH_2$ and $iLH_2$, which can be attributed to the sensitivity of the photon counting capabilities of the CCD cameras fitted in optical imagers (*Cool et al., 2013*). The crystal structure reported here will be important for further optimisation of this near-infrared dual BLI system, particularly to increase the brightness of the current enzymes with $iLH_2$, as well as the discovery of novel luciferase colour-mutants which could be used for multi-coloured BLI in the near-infrared. Additionally, the use of this system in combination with luciferases utilising other substrates for multi-coloured BLI could also be explored, as well as its application to monitoring more complex processes in animal models (*Kleinovink et al., 2018*). This work represents an important step forward in increasing the utility of BLI and opens up the window for multi-coloured BLI in the near-infrared.

## Materials and methods

### Key resources table

| Reagent type (species) or resource | Designation | Source or reference | Identifiers | Additional information |
|---|---|---|---|---|
| Gene (*Photinus pyralis*) | FLuc | this paper | N/A | Amino acid changes from ref 21 and 23, codon optimised for mammalian expression for use in this paper |

*Continued on next page*

*Continued*

| Reagent type (species) or resource | Designation | Source or reference | Identifiers | Additional information |
|---|---|---|---|---|
| Strain, strain background (mouse, male) | NSG | Jax mouse strain (*Charles River*) | NOD.Cg-*Prkdcscid Il2rgtm1Wjl/SzJ* | N/A |
| Cell line (Human, male) | Raji B lymphoma | ATCC CCL-86 | N/A | Mycoplasma tested by GATC (Eurofins Genomics) |
| Antibody | Anti-human CD34-PE | Biolegend | Clone 581, RRID: AB_1731862 | (1:20) |
| Antibody | Anti-human CD271-APC | Biolegend | Clone ME20.4, RRID: AB_10645515 | (1:20) |
| Chemical compound, dye | Viability APC eFluoro780 | eBioscience | N/A | (1:1000) |
| Antibody | Anti-mouse/human CD11b PerCP/Cy5.5 | Biolegend | Clone M1/70, RRID: AB_893232 | (1:20) |
| Antibody | Anti-human CD19 FITC | eBioscience | Clone HIB19, RRID: AB_10669461 | (1:20) |
| Antibody | Anti-human CD20 eFLuor 450 | eBioscience | Clone 2H7, RRID: AB_1633384 | (1:20) |
| Chemical compound, drug | Luciferin | Regis technologies | N/A | N/A |
| Software, algorithm | Living Image | Perkin Elmer | N/A | N/A |
| Software, algorithm | Prism | Graphpad | N/A | N/A |
| Software, algorithm | Excel | Microsoft | N/A | N/A |
| Software, algorithm | Flow Jo | Tree Star Inc (Oregon, USA) | N/A | N/A |
| Software, algorithm | PyMOL software | Schrodinger | N/A | N/A |
| Software, algorithm | CrysalisPro | Agilent Technologies | N/A | N/A |
| Software, algorithm | BD FACSDIVA | BD biosciences | N/A | N/A |
| Other | | | | |
| Chemical compound, drug | infraluciferin | Anderson, J.C.; Grounds, H.; Jathoul, A.P.; Murray, J.A.H.; Pacman, S.J.; Tisi, L. RSC Advances 2017, 7, 3975–82 | N/A | Prepared by JC Anderson laboratory |
| Chemical compound, drug | iDLSA | this paper | N/A | Prepared by JC Anderson laboratory, see below and data set at https://doi.org/10.5061/dryad.3j9kd51cs. |
| Chemical compound, drug | CycLuc1 | Merck Millipore | N/A | N/A |
| Chemical compound, drug | Aka-Lumine-HCL | Wako Pure Chemical Industries | N/A | N/A |

## Preparation of 5'-o-[(n-dehydroinfraluciferyl)-sulfamoyl] adenosine dehydroinfraluciferin

All manipulations were routinely carried out under an inert (Ar or $N_2$) atmosphere. All reagents were used as received unless stated. For the purposes of thin layer chromatography (tlc), Merck silica-aluminium plates were used, with uv light (254 nm) and potassium permanganate or anisaldehyde for visualisation. For column chromatography Merck Geduran Si 60 silica gel was used. Butyl lithium solutions were standardised with diphenyl acetic acid.

Melting points are uncorrected and were recorded on a Griffin melting point machine. Infrared spectra were recorded using a Bruker Alpha ATR spectrometer. All NMR data were collected using a Bruker AMX 300 MHz or Bruker AVANCE III 600 MHz as specified. Reference values for residual solvents were taken as $\delta$ = 7.27 ($CDCl_3$), 2.51 (DMSO –$d6$), 3.30 (MeOD- $d4$) ppm for [1]H NMR and $\delta$ = 77.2 ($CDCl_3$), 39.5 (DMSO –$d6$), 49.0 (MeOD- $d4$) ppm for [13]C NMR. [19]F NMR spectra were measured using a Bruker DX300 spectrometer, referenced to trichlorofluoromethane. Coupling constants (J) are given in Hz and are uncorrected. Where appropriate COSY and DEPT experiments were carried out to aid assignments. Mass spectroscopy data were collected on a Micromass LCT Premier XE (ESI) instrument. Elemental analysis was performed on an Exeter Analytical Inc CE-440 CHN analyser.

6-($\beta$-Methoxyethoxymethylether)benzothiazole (**1**), (*Muramoto et al., 1999*)

6-($\beta$-Methoxyethoxymethyl ether)−2-formylbenzothiazole (**2**), (*Anderson et al., 2017*)

1-(4-methoxycarbonylthiazole)methyltriphenylphosphonium chloride (*Hermitage et al., 2001*; *Old, 2008*)

2′,3′-O-Isopropylidene-5′-O-sulfamoyladenosine (*Heacock et al., 1996*) were synthesised using procedures reported in the literature.

6-($\beta$-Methoxyethoxymethoxy)−2-(2-(4-methoxycarbonylthiazol-2-yl)ethenyl) benzothiazole (**3**). A suspension of aldehyde **2** (200 mg, 0.748 mmol) and 1-(4-methoxycarbonylthiazole)methyltriphenylphosphonium chloride (680 mg, 1.50 mmol) in DMF (3.5 mL) was cooled to 0℃ and treated with $K_2CO_3$ (348 mg, 2.52 mmol). The resultant solution was allowed to warm to rt and stirred for 16 hr. After this time $H_2O$ (40 mL) was added and the solution extracted using EtOAc (2 × 40 mL). The organics were combined and washed with $H_2O$ (40 mL), separated, dried ($MgSO_4$), filtered and concentrated *in vacuo*. Purification was achieved using flash column chromatography (60% $Et_2O$/Hexane) to give three as a mixture of *cis* and *trans* isomers (163 mg, 54%).

*Cis*: $R_f$ = 0.52 (60 % EtOAc/Hexane); [1]H NMR (600 MHz, $CDCl_3$) $\delta$ 3.39 (3H, s, OC$H_3$), 3.58–3.59 (2H, m, OC$H_2$C$H_2$O), 3.86–3.89 (2H, m, OC$H_2$C$H_2$O), 4.00 (3H, s, OC$H_3$), 5.36 (2H, s, OC$H_2$O), 6.96 (1H, d, J = 12.8, C$H$C(N)S), 7.25–7.29 (2H, m, Ar$H$, C$H$C(N)S), 7.62 (1H, d, J = 2.3, Ar$H$), 8.06 (1H, d, J = 8.9, Ar$H$), 8.28 (1H, s, C$H$S). In $CDCl_3$ solution the *trans* isomer was seen to isomerise to the *cis* isomer.

*Trans*: A pure sample of *trans* was by separated by column chromatography to give three as a yellow solid. m.p. 112–115℃; $R_f$ = 0.48 (60 % EtOAc/Hexane); IR $\nu_{max}$ 3101, 2926 ($\nu_{CH}$), 2889 ($\nu_{CH}$), 2818 ($\nu_{CH}$), 1716 ($\nu_{CO}$), 1629, 1598, 1556, 1495, 1456, 1333, 1318, 1281, 1239, 1222, 1208, 1162, 1089, 1046, 978 cm$^{-1}$; [1]H NMR (600 MHz, $CDCl_3$) $\delta$ 3.39 (3H, s, OC$H_3$), 3.58–3.59 (2H, m, OC$H_2$C$H_2$O), 3.86–3.88 (2H, m, OC$H_2$C$H_2$O), 4.00 (3H, s, OC$H_3$), 5.35 (2H, s, OC$H_2$O), 7.22 (1H, dd, J = 8.9, 2.4, Ar$H$), 7.59 (1H, d, J = 2.4, Ar$H$), 7.68 (2H, s, C$H$C(N)S), 7.92 (1H, d, J = 8.9, Ar$H$), 8.21 (1H, s, C$H$S); [13]C NMR (150 MHz, $CDCl_3$) $\delta$ 52.8 ($CH_3$), 59.2 ($CH_3$), 68.0 ($CH_2$), 71.7 ($CH_2$), 94.0 ($CH_2$), 107.5 (CH), 117.9 (CH), 124.1 (CH), 128.2 (CH), 128.7 (CH), 136.2 (C), 148.1 (C), 148.9 (C), 156.3 (C), 161.7 (C), 162.9 (C), 165.6 (C); m/z (ESI) 407 (100%, M$^+$+H); HRMS $C_{18}H_{19}N_2O_5S_2$ calcd. 407.0735, found 407.0738.

(E)−6-($\beta$-Methoxyethoxymethoxy)−2-(2-(4-carboxythiazol-2-yl)ethenyl) benzothiazole (**4**). A suspension of **3** (20 mg, 0.049 mmol) in THF (0.75 mL) and $H_2O$ (0.37 mL) was treated with $LiOH.H_2O$ (5.0 mg, 0.12 mmol) and stirred for 15 min. After this time $H_2O$ (10 mL) and EtOAc (10 mL) were added and the layers separated. The aqueous layer was acidified with 2 M HCl and extracted using EtOAc (2 × 10 mL), organics dried over $MgSO_4$, filtered and concentrated *in vacuo* to give **4** (19 mg, quant.) as a yellow solid. m.p. 175–178℃; $R_f$ = 0.20 (50 % EtOAc/MeOH); IR $\nu_{max}$3101 ($\nu_{OH}$), 2917 ($\nu_{CH}$), 1680 ($\nu_{CO}$), 1598, 1554, 1455, 1398, 1320, 1241, 1204, 1160, 1101, 1045, 938 cm$^{-1}$; [1]H NMR (600 MHz, MeOD-$d4$) $\delta$ 3.33 (3H, s, OC$H_3$), 3.57–3.58 (2H, m, OC$H_2$C$H_2$O), 3.84–3.85 (2H, m, OC$H_2$C$H_2$O), 5.36 (2H, s, OC$H_2$O), 7.25 (1H, dd, J = 8.9, 2.4, Ar$H$), 7.69 (1H, d, J = 2.4, Ar$H$), 7.71

(1H, dd, $J$ = 16.1, 0.6, C$H$C(N)S), 7.73 (1H, d, $J$ = 16.1, C$H$C(N)S), 7.80 (1H, d, $J$ = 8.9, Ar$H$), 8.43 (1H, s, $J$ = 7.5, C$H$S); $^{13}$C NMR (150 MHz, MeOD-$d$4) δ 59.1 (CH$_3$), 69.0 (CH$_2$), 72.8 (CH$_2$), 95.0 (CH$_2$), 108.7 (CH), 119.0 (CH), 124.7 (CH), 128.5 (CH), 129.1 (CH), 130.1 (CH), 137.5 (C), 149.8 (C), 150.0 (C), 157.7 (C), 163.8 (C), 164.6 (C), 166.7 (C); m/z (ESI) 393 (100%, M$^+$+H), 300 (9%); HRMS C$_{17}$H$_{17}$N$_2$O$_5$S$_2$ calcd. 393.0579, found 393.0581; Anal. Calcd. for C$_{17}$H$_{16}$N$_2$O$_5$S$_2$: C, 52.03; H, 4.11; N, 7.14. Found C, 51.85; H, 4.29; N, 6.71%.

6-($β$-Methoxyethoxymethoxy)−2-(2-(4-pentafluorophenoxycarbonylthiazol-2-yl)ethenyl) benzothiazole (**5**). A solution of **4** (30 mg, 0.077 mmol) in pyridine (3.80 mL) was treated with EDC (18 mg, 0.096 mmol) and pentafluorophenol (18 mg, 0.096 mmol) and stirred at rt for 16 hr. The reaction mixture was concentrated *in vacuo*. Purification was achieved using flash column chromatography (30% Et$_2$O/hexane) to give five as a yellow solid (37 mg, 86%). m.p. 87–91°C; R$_f$ = 0.25 (30 % EtOAc/ Pet. Ether); IR ν$_{max}$ 2922, 2887, 2835, 1764 (ν$_{CO}$), 1599, 1554, 1486, 1470, 1454, 1321, 1290, 1260, 1242, 1211, 1199, 1180, 1137, 1123, 1100, 1060, 1010, 986 cm$^{-1}$; $^1$H NMR (600 MHz, CDCl$_3$) δ 3.40 (3H, s, OC$H_3$), 3.59–3.60 (2H, m, OC$H_2$CH$_2$O), 3.87–3.88 (2H, m, OC$H_2$CH$_2$O), 5.36 (2H, s, OC$H_2$O), 7.23 (1H, dd, $J$ = 8.9, 2.4, Ar$H$), 7.60 (1H, d, $J$ = 2.4, Ar$H$), 7.72 (1H, dd, $J$ = 16.2, 0.6, C$H$C(N)S), 7.74 (1H, d, $J$ = 16.2, C$H$C(N)S), 7.94 (1H, d, $J$ = 9.0, Ar$H$), 8.40 (1H, d, $J$ = 0.4, C$H$S); $^{13}$C NMR (150 MHz, CDCl$_3$) δ 59.2 (CH$_3$), 68.0 (CH$_2$), 71.7 (CH$_2$), 94.0 (CH$_2$), 107.5 (CH), 117.9 (CH), 124.3 (CH), 124.9 (C), 127.3 (CH), 129.8 (CH), 131.4 (CH), 136.4 (C), 137.3 (C), 139.0 (C), 140.6 (C), 142.2 (C), 144.7 (C), 149.2 (C), 156.4 (C), 156.8 (C), 162.5 (C), 166.6 (C); $^{19}$F NMR (CDCl$_3$, 282) δ −150.0 (d, $J$ = 16.9, Ar$F$), −157.14 (app t, $J$ = 19.7, Ar$F$), −161.9 (dd, $J$ = 19.7, 16.9, Ar$F$); m/z (ESI) 559 (100%, M$^+$+H); HRMS C$_{23}$H$_{15}$F$_5$N$_2$O$_5$S$_2$ calcd. 559.0415, found 559.0418.

2′,3′-$O$-Isopropylidene-5′-$O$-[$N$-(6-($β$-methoxyethoxymethoxy)-dehydroinfraluciferyl)-sulfamoyl] adenosine (**6**). A solution of 2′,3′-$O$-Isopropylidene-5′-$O$-sulfamoyladenosine (20 mg, 0.052 mmol) in DMF (1.8 mL) was treated with DBU (11 mg, 0.076 mmol) and stirred at rt for 10 min. A solution of **5** (29 mg, 0.052 mmol) in DMF (0.2 mL) was then added dropwise. The reaction was stirred at rt for 16 hr. After this time pyridine (0.15 mL) was added and the solution stirred for 4 hr. The resultant solution was concentrated *in vacuo* and purified using flash column chromatography (5% MeOH/DCM) to give **6** (30 mg, 76%) as a yellow solid. m.p. 164°C, dec.; R$_f$ = 0.32 (5 % MeOH/DCM); IR ν$_{max}$3290 (ν$_{OH}$), 2932 (ν$_{CH}$), 1644 (ν$_{CO}$), 1598, 1552, 1505, 1457, 1418, 1373, 1291, 1250, 1209, 1150, 1103, 1080, 1048, 985 cm$^{-1}$; $^1$H NMR (600 MHz, DMSO-$d$6) δ 1.34 (3H, s, C(C$H_3$)$_2$), 1.54 (3H, s, C(C$H_3$)$_2$), 3.21 (3H, s, OC$H_3$), 3.46–3.49 (2H, m, OC$H_2$CH$_2$O), 3.74–3.78 (2H, m, OC$H_2$CH$_2$O), 4.09 (1H, dd, $J$ = 11.0, 4.9, C$H_2$OS(O)$_2$), 4.12 (1H, dd, $J$ = 11.0, 4.9, C$H_2$OS(O)$_2$), 4.42–4.46 (1H, m, OC$H$CH$_2$), 5.09 (1H, dd, $J$ = 6.1, 2.5, C$H$CHO), 5.36 (2H, s, OC$H_2$O), 5.41 (1H, dd, $J$ = 6.1, 2.9, C$H$CHNO), 6.17 (1H, d, $J$ = 2.9, CHC$H$NO), 7.22 (1H, dd, $J$ = 8.9, 2.5, Ar$H$), 7.35 (2H, br, N$H_2$), 7.71 (1H, d, $J$ = 16.1, C$H$C(N)S), 7.79 (1H, d, $J$ = 16.0, C$H$C(N)S), 7.79 (1H, d, $J$ = 2.5, Ar$H$), 7.95 (1H, d, $J$ = 8.9, Ar$H$), 8.11 (1H, s, C$H$S), 8.12 (1H, s, NC$H$(N)), 8.44 (1H, s, NC$H$(N)); $^{13}$C NMR (150 MHz, DMSO-$d$6) δ 25.2 (CH$_3$), 27.1 (CH3), 58.1 (CH$_3$), 67.3 (CH$_2$), 67.6 (CH$_2$), 71.0 (CH$_2$), 81.6 (CH), 83.5 (CH), 83.9 (CH), 89.3 (CH), 93.3 (CH2), 108.0 (CH), 113.2 (C), 117.4 (CH), 118.8 (C), 123.6 (CH), 125.2 (CH), 126.1 (CH), 128.5 (CH), 136.0 (C), 139.6 (CH), 148.6 (C), 149.0 (C), 152.8 (CH), 155.3 (C), 156.1 (C), 156.8 (C), 162.8 (C), 162.9 (C), 165.0 (C); m/z (ES$^+$) 761 (100%, M$^+$+H); HRMS C$_{30}$H$_{30}$N$_8$O$_{10}$S$_3$ calcd. 761.1482, found 761.1486.

6-Hydroxy-2-(4-1E,3E-(4-ethoxycarbonyl-4,5-dihydrothiazol-2-yl)buta-2,4-dienyl)benzothiazole (iDLSA). A solution of **6** (20 mg, 0.026 mmol) in TFA (0.32 mL) was stirred at rt for 2 hr and then H$_2$O (0.1 mL) added and the solution concentrated *in vacuo*. EtOH (2 mL) added and concentrated *in vacuo* to give the TFA salt of iDLSA (18 mg, 94%) as an orange solid. m.p. 68–70°C; IR ν$_{max}$3102 (ν$_{OH}$), 1667 (ν$_{CO}$), 1426, 1132 cm$^{-1}$; $^1$H NMR (600 MHz, DMSO-$d$6) δ 4.20–4.25 (2H, m, C$H$OH, C$H$OH), 4.53–4.59 (3H, m, C$H$OC, C$H_2$OS(O)$_2$), 5.96 (1H, d, $J$ = 5.0, CHC$H$NO), 7.00 (1H, d, $J$ = 2.4, Ar$H$), 7.42 (1H, d, $J$ = 2.5, Ar$H$), 7.72 (1H, d, $J$ = 15.9, C$H$C(N)S), 7.84 (1H, d, $J$ = 8.8, Ar$H$), 7.95 (1H, d, $J$ = 15.8, C$H$C(N)S), 8.33 (1H, s, C$H$S), 8.53 (1H, s, NC$H$(N)), 8.59 (1H, s, NC$H$(N)). $^{13}$C NMR too weak due to poor solubility of compound. MS did not give M$^+$ or meaningful fragment for accurate mass measurement.

Copies of all 1H and 13C NMR have been deposited at https://doi.org/10.5061/dryad.3j9kd51cs.

## Preparation of the p.pyralis luciferase-5'-o-[(n-dehydroinfraluciferyl)-sulfamoyl] adenosine dehydroinfraluciferin (FLuc-iDLSA) complex

Approximately 0.6 mg of iDLSA was suspended in 500 µL of crystallisation buffer (25 mM Tris-Cl containing 200 mM AmSO4, 1 mM DTT, 1 mM EDTA) pH 7.85 at 21°C. The solution was vortexed vigorously and sonicated. Most of the solid was dissolved and the concentration was determined to be ~1 mM by UV absorbance (using an extinction coefficient of 8200 at 372 nm for this buffer and pH). A 20 mg/mL solution of *P. pyralis* luciferase (PpyWT that contains the N-terminal peptide GPLGS-) in the same buffer (500 µL) was mixed gently with the iDLSA at room temperature and then incubated at 15°C for 20 min. The concentration of iDLSA in the protein-inhibitor mixture was determined by UV absorbance to be 620 µM, giving an inhibitor:enzyme ratio of ~3:1, at this point the bioluminescence activity of the mixture was assayed and the enzyme was 85% inhibited. A small amount of a separate iDLSA solution (available from solubility trials) was added to bring the inhibitor:enzyme ratio to ~4:1, and based on activity the enzyme was 91% inhibited. Finally, the protein-inhibitor solution was added to ~0.5 mg of iDLSA and mixed gently and incubated at 15°C for 15 min. Based on activity, 99% of the enzyme was inhibited and based on UV absorbance the inhibitor:enzyme ratio was 5.6:1 (950 µM:170 µM) The protein-inhibitor solution was centrifuged and a very slight amount of inhibitor was evident. The supernatant was frozen in liquid nitrogen in ~18–50 µL aliquots and stored at −80°C. A single aliquot was thawed and the solution remained clear. The pH of this solution at 6°C should be 8.3, pH 8.17 at 10°C, and pH 7.9 at 21°C.

## Crystallisation and refinement of P. pyralis luciferase/dehydroinfraluciferin DLSA complex

Approximately 0.6 mg of iDLSA was resuspended in 500 µL of buffer (25 mM Tris-Cl containing 200 mM $(NH_4)_2SO_4$, 1 mM DTT, 1 mM EDTA) pH 7.85 at 21°C to a concentration of 1 mM as determined by UV absorbance (extinction coefficient of 8200 at 372 nm). This solution was mixed with a 20 mg/mL solution of *P. pyralis* (inhibitor:enzyme ratio of ~3:1) at room temperature and then incubated at 15°C for 20 min. The inhibitor:enzyme solution was centrifuged and the supernatant was frozen in liquid nitrogen in ~18–50 µL aliquots and stored at −80°C for future crystallisations.

Crystallisations used the hanging drop vapour diffusion method. Drops containing 1–2 µl of inhibitor:enzyme solution were mixed with the same volume of well solution and equilibrated against 500 µl of well solution, incubated at 4°C, with crystals typically appearing within 48 hr. Glycerol was used as a cryoprotectant, in an optimised well solution of 150 mM $(NH_4)_2SO_4$, 50 mM HEPES pH 7.0, 2% PEG 1000.

Data were collected at the Diamond Light Source on beam line IO4-1, at wavelength 0.91587 Å, and 100 K. Processing and data reduction were carried out on site using CrysalisPro (*Agilent Technologies*), and synchrotron data sets were processed and scaled by using XDS, SCALA and XIA2 programs. Molecular replacement methods were used successfully to determine the relative orientation and position of the two monomers in the asymmetric unit using the PHASER program (*McCoy et al., 2007*). The starting dehydroinfraluciferin DLSA complex model was derived from the Firefly luciferase apo structure (PDB-ID 3IEP) with all solvent atoms and the luciferin removed. A simple rigid body refinement was sufficient to initiate refinement, with subsequent refinement and model building cycles performed using Refmac5 and Coot (*Murshudov et al., 2011*; *Emsley et al., 2010*).

The X-ray data collection and refinement statistics have been deposited at https://doi.org/10.5061/dryad.3j9kd51cs.

## Firefly luciferase mutants and cell lines

FLuc mutants contained 11 pH and temperature stabilising mutations (F14R/L35Q/A105V/V182K/T214C/I232K/D234G/E354R/D357Y/S420T/F465R) (*Jathoul, 2012*). FLuc_green contained an additional three mutations (V241I/G246A/F250S), and FLuc_red has the red-shifting mutation S284T, as well as the mutation R354I which is required to maintain the red-shift in this stabilised FLuc backbone (*Branchini et al., 2007*). All FLuc mutants were codon optimised for mammalian expression and cloned into the MLV-based splicing gamma retroviral vector SFG. The Raji B lymphoma cell line used in all experiments was transduced to express a FLuc mutant, and subsequently flow-sorted for pure FLuc expressing populations using a co-expressed marker gene. For tumour cell lines FLuc.IRES

was upstream of the marker gene CD34 or dNGFR as indicated. For T cells FLuc.2A_peptide was upstream of the CAR CD19-4G7_HL-CD8STK-41BBZ.

## Production of retroviral supernatant

HEK-293T packaging cells were plated at a density of 200'000 cells/ml in 100 mm tissue culture dish ~24 hr prior to transfection. Transfections were performed when cells were 50–70% confluent. A bulk transfection mixture was prepared where 30 μl GeneJuice Transfection Reagent (*Merck millipore*) was added to 470 μl of plain RPMI for each supernatant to be produced. Following a 5 min incubation at room temperature, a total volume of 12.5 μg of DNA was added for each plate to be transfected (for retroviral transfection: 3.125 μg RDF RD114 env plasmid, 4.6875 μg PeqPam-env gagpol plasmid, 4.6875 μg SFG retroviral construct). Following addition of plasmid DNA, the mixture was incubated for a further 15 min at room temperature prior to dropwise addition to the HEK-293 T cell culture. Plates were gently agitated following transfection. Supernatant harvested at 48 hr was stored at 4°C, and was then combined with the 72 hr harvest prior to aliquoting and storage at −80°C.

## Transduction of cell lines

The day prior to transduction Raji B lymphoma cells (atcc ccl-86) (>90% viable) were diluted ~1 in 10 to ensure exponential growth for transduction; also a well of non-tissue culture treated 24 well plate was coated with 8 μg/ml retronectin (*Lonza*) for every plasmid to be transduced and left at 4°C overnight. The next day retronectin was aspirated and 250 μl of each retroviral supernatant for transduction was added to a well and incubated for 30 min at room temperature. Whilst incubating, Raji cells were harvested, counted and resuspended at a concentration of 600,000 cells/ml. supernatants were aspirated from wells of retronectin coated plate and 500 μl of cell suspension was added to each well followed by 1.5 ml of the same retroviral supernatant that was previously incubated in each well. Cells were spin transduced at 1000 RCF for 40 min then returned to incubator for 48 hr before harvest and expression testing.

## Flow cytometry and Fluorescence Activated Cell Sorting

Transduction efficiencies were assessed by flow cytometry, based on marker gene expression as indicated by antibody staining using the BD LSR FortessaX-20. If necessary Fluorescence Activated Cell Sorting (FACS) was performed to obtain pure expressing populations, also based on marker gene expression as indicated by antibody staining, using the BD FACS Aria Fusion. FACS was also use to sort populations of cells expressing differing levels of FLuc_green and FLuc_red within the same cell. Concentration of antibody used was guided by manufacturer's instructions. Anti-human CD34-PE (clone 581), anti-human CD271-APC (clone ME20.4) and anti-mouse/human CD11B-PerCP/Cy5.5 (clone M1/70) (*Biolegend*). Anti-humanCD19-FITC (clone HIB19), anti-human CD20-eFluor450 (clone 2H7) and Viability APC-eFluor780 (*eBioscience*). Data were analysed using Flow Jo software (*Tree Star Inc, Oregon, USA*).

## Flow cytometry of extracted bone marrow

When bone marrow cells were required for flow cytometry analysis. Following animal sacrifice by $CO_2$ narcosis and cervical dislocation, the femurs were removed and transferred to PBS pending cellular harvest. The ends of the femur were snipped off using scissors and the bone was placed in an extraction tube (microfuge tube with holder made from 200 μl pipette tip inserted). Tubes were centrifuged at 1000 RCF for 60 s. Bone marrow pellet was resuspended in 50 μl Ammonium-Chloride-Potassium (ACK) lysing buffer (*Lonza*) and left for 60 s at room temperature before washing with PBS and passing through a 70 μm filter before pelleting. Samples were blocked 2.4G2 supernatant (rat anti-mouse CD32) supplemented with mouse Ig FcR blocking reagent (*Miltenyl Biotec*) for 30 min at room temperature. Cells were washed with PBS and pelleted, followed by each sample being transferred to a well of a U-bottomed 96 well plate before proceeding with antibody staining. An antibody master mix containing all antibodies to be used for staining was prepared in PBS to a total volume of 100 μl per sample. Samples were left to stain at room temperature in the dark for 30 min. Samples were washed once with PBS, pelleted and transferred to FACS tubes. Beckman Coulter Flow-Checkfluorospheres were used as a stopping gate for flow cytometry analysis. Beads are

supplied at $1 \times 10e^6$ beads/ml in an aqueous solution containing preservative surfactant. To prevent toxicity to cellular samples, beads were washed once with PBS prior to addition to samples. Following centrifugation (400 RCF for 5 min), beads were resuspended in an equal volume of PBS with 10 µl of beads added to each sample.

As six fluorophores were used for bone marrow analysis, compensation was performed prior to sample acquisition using OneComp eBeads (*eBioscience*). Events were kept between 2,000–5,000 events/second, with 1000 events being recorded per sample, using flow check beads as a stopping gates (10% each sample). Flow cytometry gating first identified the lymphocyte population (FSC-A vs SSC-A), exclusion of doublet cells (SSC-A vs SSC-W); antibodies detailed in **Flow cytometry and Fluorescence Activated Cell Sorting** were then used to gate on viable cells, exclude mouse monocyte cells (mCD11b), identify the Raji tumour cell population (CD19 and CD20) and finally co-expressed marker gene (dNGFR and dCD34).

## Preparation of CAR T cells

On day one peripheral blood mononuclear cells were isolated from a healthy donor blood using Ficoll-paque density gradient media (*GE Healthcare*). Cells were resuspended at $2 \times 10^6$/ ml and stimulated with 1 mg/ml PHA (*Sigma*). On day 2 cells were fed with IL-2 at a concentration of 100 U/ ml (*Genscript*). On day 3 cells were transduced as described in **transduction of cell lines**, with IL-2 at a final concentration of 100 U/ml. On day 6 cells were harvested and resuspended at $1 \times 10^6$/ml with 100 U/ml IL-2 and left to recover for at least 48 hr before in vivo injection. Transduction efficiency was measured using flow cytometry.

## In vitro bioluminescence assays

For spectrographic testing of FLuc, mutants were stably transduced in the mammalian Raji B-cell lymphoma cell line. For in vitro bioluminescence assays cells were harvested, counted and $1 \times 10^6$ cells/well were resuspended in TEM buffer (1M Tris-acetate, 20 mM EDTA and 100 mM MgSO4 at pH 7.8) and added in triplicate to wells of a black 96-well plate (100 µl/well). If mixtures of cells were used, total cell number remained the same. For spectral testing the stage temperature of the IVIS Spectrum was set to 37°C (automatic acquisition mode, FOV 13.2, f/1). $iLH_2$ was synthesised by UCL Chemistry. Other substrates tested include D-luciferin (*Regis Technologies*), CycLuc1 (*Merck Millipore*) and Aka-Lumine-HCL (*Wako Pure Chemical Industries*). Substrates were dispensed into the wells using a multi-channel pipette (at a final concentration of 300 µM). A 2 min delay was allowed for stabilisation of light output. Images were acquired through all 18 bandpass filters on the IVIS Spectrum (20 nm bandpass, 490 nm to 840 nm). Living image software (*Perkin Elmer*) was used for ROI analysis of spectral images and spectral unmixing analysis. Image analysis involved placing a ROI over the signal in each well. If a series of spectral images was acquired, the same ROI was placed over the well in every image for each plate. For spectral unmixing analysis, guided spectral unmixing was first used on pure expressing FLuc_green and FLuc_red populations to create a library spectra for each mutant with each substrate. The relevant library spectra was then used to perform spectral unmixing on mixed FLuc_green and FLuc_red populations (or cellular populations expressing both enzymes). Data exported to Excel (*Microsoft*) and Prism (*Graphpad*) for further analysis. Spectra was normalised to peak emission for each FLuc mutant with each substrate. Due to the characterisation nature of these in vitro experiments, and the substantial amounts of precious chemicals needed to synthesise $iLH_2$, it was decided to repeat each in vitro experiment twice with three replicates.

## In vivo models

All animal studies were approved by the University College London Biological Services Ethical Review Committee and licensed under the UK Home Office regulations and the Guidance for the Operation of Animals (Scientific Procedures) Act 1986 (Home Office, London, United Kingdom). All of the in vivo models used the severely immunocompromised NSG (NOD.Cg *Prkdc*$^{scid}$*Il2rg*$^{tm1Wji}$/SzJ) mouse model (JAX mouse strain, *Charles River*). Mice were male and aged between 6–8 weeks old. Due to the characterisation, or proof of concept, nature of these experiments, and the substantial amounts of precious chemicals needed to synthesise iLH$_2$, it was decided to engraft 4–5 mice for every condition in each model to ensure engraftment and survival in a least three animals for each

condition. Also, no specific toxicity experiments were performed, but no adverse side effects were observed with iLH$_2$.

For engraftment of subcutaneous tumours, FLuc expressing Raji cell lines were counted and $2 \times 10^6$ cells were pelleted for each animal to be injected. Cells were washed twice in PBS before being resuspended in plain RPMI 1% HEPES (*Sigma-Aldrich*) to a concentration of $2 \times 10^7$ cells/ml and were kept on ice ready for injection. Cells were injected subcutaneously (100 µl bolus) into a shaved area of the flank. Mice were left at least 5 days for tumour development before imaging.

For engraftment of systemic tumours, FLuc expressing Raji cell lines were counted and $5 \times 10^5$ cells were pelleted for each animal. If mixtures of two different FLuc expressing cell lines were used, total cell number per mouse remained at $5 \times 10^5$ cells. Cells were washed twice in PBS before being resuspended in plain RPMI 1% HEPES to a concentration of $2.5 \times 10^6$ cells/ml and were kept on ice ready for injection. Animals were transferred to a warming chamber set at 39–42°C to facilitate peripheral vasodilation prior to intravenous (IV) injections. Mice were placed in a restraint and cells were injected IV (200 µl bolus) into the tail vein. Mice were left at least 7 days for tumour development before imaging. For the CAR T cell model $5 \times 10^6$ CAR positive T cells were injected IV (200 µl bolus) 8 days after Raji cell engraftment.

For engraftment of intracranial tumours, FLuc expressing Raji cell lines were counted and $2 \times 10^4$ cells were pelleted for each animal. Cells were washed twice in PBS before being resuspended in PBS to a concentration of $1 \times 10^4$ cells/µl and were kept on ice ready for injection. Intracranial injections were performed using a stereotaxic frame fitted with a hamilton syringe. Cells were injected (2 µl bolus) into the right striatum (from bregma 2 mm right, 1 mm anterior, 4 mm down). Mice were left at least 7 days for tumour development before imaging.

## In vivo bioluminescence imaging

For imaging of in vivo models, LH2 and iLH2 were solubilised in sterile PBS and animals were administered with substrate (2 mg (or 100 mg/kg) of either LH$_2$ or iLH$_2$ in 200 µl or 400 µl bolus respectively) via intraperitoneal (IP) injection. Animals were anaesthetised using 2% Isofluorine (flow rate 1 L/min O$_2$). Spectral imaging was commenced 10 min post IP injection to allow stabilisation of light output. If the same animal was being imaged with both LH2 and iLH2, at least 24 hr was left between imaging to allow for full clearance of substrate. In vivo bioluminescent images were acquired using IVIS Spectrum (FOV 24, f/1, Medium (8)bin, automatic acquisition mode for imaging with LH$_2$, FOV 24, f/1, Medium (8)bin, 120 s acquisition, total imaging time 24 mins for imaging with iLH$_2$). These parameters are calculated to keep the binning, exposure time and f/stop within an optimal range for quantification. Up to five animals could be imaged at once and the stage was heated to 37°C. Open filter images were acquired prior to and post spectral imaging to confirm the stability of photon emission during spectral acquisition. Spectral imaging acquired images through 14 and 12 of the 20 nm bandpass filters on the IVIS Spectrum depending on substrate used (530–830 nm for LH2 and 590–830 nm for iLH2), starting from the lowest to the highest filter. It was not necessary to acquire images through all filters as the bioluminescent emissions of FLuc mutants did not cover the full spectral range from 490 to 850 nm. Living image software was used for ROI analysis of spectral images and spectral unmixing analysis. Radiance values for bioluminescence are shown using pseudo-colour scales detailed in each image. Image analysis involved placing an ROI over the tumour signal for every animal in each model. If a series if spectral images were acquired, the same ROI was placed over tumour signal in every image for each mouse. For spectral unmixing analysis, guided spectral unmixing was first used on pure expressing FLuc_green and FLuc_red populations from spectral characterisation experiments to create a library spectra for each mutant with each substrate. The relevant library spectra was then used to perform spectral unmixing on mixed FLuc_green and FLuc_red populations. Data exported to Excel (*Microsoft*) and Prism (*Graphpad*) for further analysis.

## Statistical analysis

Where relevant means ± standard deviation of data given. Statistical tests used include T test and ONE-way ANOVA with post hoc Tukey's test for multiple column comparison (*Prism, Graphpad*). Correlation analysis performed using *Microsoft excel*.

## Acknowledgements

We would like to thank Steven Pacman for the synthesis of infraluciferin, and the following masters students for their contributions to the synthesis of iDLSA, Laszlo Berencei (EPSRC vacation bursary) Boyuan Deng and Julia Holm (Erasmus), the MRC based at Imperial College London for the use of their IVIS Spectrum, especially to Dr Alex Sardini, and the core flow cytometry facility at UCL Cancer Institute for the sorting of cell lines. This work was partly funded by the National Science Foundation grant MCB-1410390, Force Office of Scientific Research grant FA9550-18-1-0017, Erba Diagnostics Mannheim and the EPSRC for an industrial CASE award EP/L504889/1 (SJP) and UCL.

## Additional information

### Funding

| Funder | Grant reference number | Author |
|---|---|---|
| National Science Foundation | MCB-1410390 | Tara L Southworth<br>Bruce R Branchini |
| Air Force Office of Scientific Research | FA9550-18-1-0017 | Tara L Southworth<br>Bruce R Branchini |
| Engineering and Physical Sciences Research Council | EP/L504889/1 | Helen Allan<br>James C Anderson |
| University College London | | Helen Allan<br>James C Anderson |
| Biotechnology and Biological Sciences Research Council | | Cassandra L Stowe |

The funders had no role in study design, data collection and interpretation, or the decision to submit the work for publication.

### Author contributions

Cassandra L Stowe, Conceptualization, Data curation, Formal analysis, Investigation, Methodology, Writing—original draft, Project administration; Thomas A Burley, Tara L Southworth, Giulia Agliardi, Alastair Hotblack, Data curation, Investigation; Helen Allan, Validation, Investigation, Methodology; Maria Vinci, Investigation, Performed intracardiac injections on the mice; Gabriela Kramer-Marek, Supervision, Investigation, Project administration; Daniela M Ciobota, Investigation, Project administration; Gary N Parkinson, Conceptualization, Data curation, Formal analysis, Investigation, Methodology; Mark F Lythgoe, Supervision, Funding acquisition; Bruce R Branchini, Resources, Data curation, Formal analysis, Supervision, Investigation, Methodology, Writing—original draft, Writing—review and editing; Tammy L Kalber, Supervision, Writing—original draft, Project administration, Writing—review and editing; James C Anderson, Conceptualization, Formal analysis, Supervision, Investigation, Methodology, Writing—original draft, Project administration, Writing—review and editing; Martin A Pule, Conceptualization, Resources, Supervision, Funding acquisition, Writing—original draft, Writing—review and editing

### Author ORCIDs

James C Anderson (ID) https://orcid.org/0000-0001-8120-4125

### Ethics

Animal experimentation: All animal procedures were conducted in accordance with the Home Office Scientific Procedures Act (1986), within the guidelines of the relevant personal and project licences.

### Decision letter and Author response

Decision letter https://doi.org/10.7554/eLife.45801.021
Author response https://doi.org/10.7554/eLife.45801.022

## Additional files

### Supplementary files

• Transparent reporting form DOI: https://doi.org/10.7554/eLife.45801.015

### Data availability

All data generated or analysed during this study are included in the manuscript and data deposited at https://datadryad.org doi:10.5061/dryad.3j9kd51cs (Raw imaging data, Copies of 1H and 13C NMR of the synthetic compounds to make iDLSA, X-ray data collection and refinement statistics).

The following datasets were generated:

| Author(s) | Year | Dataset title | Dataset URL | Database and Identifier |
|---|---|---|---|---|
| Parkinson GN, Stowe C, Anderson JC | 2019 | Near-infrared dual bioluminescence imaging in vivo using infra-luciferin | http://www.rcsb.org/structure/6HPS | Protein Data Bank, 6HPS |
| James C Anderson, Martin A Pule | 2019 | Data from: Near-infrared dual bioluminescence imaging in mouse models of cancer using infraluciferin | https://doi.org/10.5061/dryad.3j9kd51cs | Dryad Digital Repository, 10.5061/dryad.3j9kd51cs |

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
