## [Decision Letter]

Thank you for submitting your article "Near-infrared dual bioluminescence imaging in vivo using infraluciferin" for consideration by *eLife*. Your article has been reviewed by two peer reviewers, and the evaluation has been overseen by a Reviewing Editor and David Ron as the Senior Editor. The following individual involved in review of your submission has agreed to reveal their identity: Laura Mezzanotte (Reviewer #1).

The reviewers have discussed the reviews with one another and the Reviewing Editor has drafted this decision to help you prepare a revised submission.

The work presented here represents an innovative approach and an advance in the field.

Essential revisions:

1) Please address if different ratios of the two enzymes expressed in one cell type be distinguished similarly to in different cells? This might be useful for gene expression studies, but would competition for substrates within a single cell cause problems?

2) Can you better explain the spectral deconvolution, its limits, and address if it’s at all possible to separate the two signals based on a ratio or two single wavelength bands. If possible, what are the limitations of this for users who only have access to systems with band pass filters. In this context, it was felt that Figure 1 could be improved to incorporate such discussion and better clarify the design and goals of the study.

3) Please discuss how some of the best aspects of other published systems might be combined with the systems described here, for example, to do three colour BLI. Along these lines, in the second paragraph of the Discussion, is there a reason the click-beetle luciferase/naphthyl luciferin system wasn't used as benchmark? This system is mentioned in the Discussion and discounted for two-color work because of a broad emission peak. It does indeed have a broad peak, however, given the success of the spectral unmixing achieved in this study, we question whether the emission peak is so broad as to discount its use for multiplexing.

4) The role of the structure in the study is not clear. The authors should better integrate the structural information or remove it to allow more of a focus on relevant information. Currently its stated that it may be valuable to improve the performance of infraluciferin, but was it of any direct use in the current work? This should be clarified.

5) The nature of the green and red mutants needs more description. It is assumed the green is the same as x11 + three additional mutations but it is not clear why this mutant was engineered. It is assumed that red is x11 + S284T + R354I (or is it E?) but it is not clear why this mutant was engineered. It is also a gap in the story that there isn't any biochemical analysis on these mutants (aside from spectra). Were they purified?

6) Its surprising that the very low photon yield from the infraluciferin system can enable collection of a useable signal. Can the authors comment on how the signal for infraluciferin is processed and if any special care was needed regarding substrate concentration/availability, camera settings, exposure time, binning or other strategies to enhance signal to noise. Along these lines, the authors used iLH_2_ concentration of 100mg/kg for their in vivo experiments. Could you go higher to enhance emission intensities? Is there toxicity data available, or would cost become a limitation?

[Editors' note: further revisions were requested prior to acceptance, as described below.]

Thank you for resubmitting your work entitled "Near-infrared dual bioluminescence imaging in mouse models of cancer using infraluciferin" for further consideration at *eLife*. Your revised article has been favorably evaluated by David Ron as the Senior Editor, a Reviewing Editor, and two reviewers.

The manuscript has been improved but there are some remaining issues that need to be addressed before acceptance, as outlined below:

The paper should be a resource for future potential users to see an exemplar, but also have the nuts and bolts information to for potential users to effectively judge that the method might work for other applications. The comments below are an effort to incorporate the necessary information to achieve this. Please read the comments carefully and work to address them in the spirit of making this a fantastic resource. But all feel it’s moving in the correct direction and this is now a minor revision that will result in a big benefit.

The reviewers have provided a set of detailed points for you to address in revision (points 1-5, the comments of reviewer 1 and points 2-3 of reviewer 1). I hope you will be able to present a revised manuscript that deals with all seven points.

Reviewer #1:

The authors have addressed the major issues raised in the preview round of review as clearly described in the rebuttal letter. The change in the title confers to the manuscript a clear focus on a specific application. However, the manuscript still require some improvement since some points need to be better clarified.

1) In the last paragraph of the subsection “Spectral characterisation of firefly luciferase mutants with LH_2_ and iLH_2_ in vivo” the authors comment that the relative intensities of the luciferases are comparable, thus the dynamic range of radiance values for both enzyme will be more similar giving a more accurate comparison of the process being monitored. The sentence results vague. For example in the case one reporter is used under the control of a constitutive promoter and the second under a control of an inducible and much weaker promoter the fact that the luciferases have similar relative intensities would not help. The authors proved that they could successfully separate in signals in cases where one reporter expression is around 10 times higher than the other (e.g. in the case of 90%/10% mixed population of cells). What happens if the expression is 99% and 1% for example? Will then the process be monitored with accuracy?

2) In the proof of concept experiment using CAR T cells, the authors show that red signal is present in "green tumor only" animals, which is probably due to leakage of green signal in the red filters even after unmixing. This demonstrates that for this application accuracy in the separation of signals may be compromised. It would be helpful if the authors explain and comment in the Results and Discussion sections how they discriminate which red signal is representative of CAR T cells and which one is not. Is it maybe just necessary to image a "green tumor only" animal together with the CAR T cells treated one? This is important and need to be clarified for making readers able to reproduce the results.

3) In the proof of concept experiments author shows an image at day 6 in which CAR T cells highly proliferate in tumors so red and green signal originate mostly in the same location in the animals. An image at earlier time point in which CAR T cells are in lymph nodes (for example 1-24 hours after injection) will allow the reader to see red and green signals in different locations in the animals. This is a minor point but will better illustrate the power of the dual color technology for cancer immunotherapy studies.

4) In the Materials and methods (subsection “in vivo bioluminescence imaging”) the authors report that imaging acquisition time is 120 sec. How long is the imaging session then using all the filters? More than 20 minutes? And is the emission of the reporters stable during this time? A comment about the length of the imaging session should be added in the Discussion.

5) In Figure 5 the percentage of red or green injected cells should be added to the unmixed images of the animals. The figure will result then clearer and of immediate interpretation.

Reviewer #2:

The authors have done an acceptable (though not ideal) job of revising this manuscript. It is in my opinion suitable for publication as a marginally impactful, yet solid report.

1) The manuscript needs editing from someone versed with the English language. For example, see incomplete sentence in Abstract (third sentence). As another example, though minor, the use of the word "origins" is awkward. The origin of luminescence emission is the photons released as a byproduct of the oxidative reaction between a luciferase and a luciferin substrate. The authors should consider an alternative word.

2) The report would benefit greatly if the authors were able to share more information about their substrate formulation, specifically for the in vivo experiments. What is the buffer? Are there any adverse effects from the substrate that the formulation was designed to address? What about solubility? Does it matter what you dissolve substrate in? In its current state I do not believe a reader could reproduce the in vivo experiments.

3) On another technical level, the final compound in the synthetic pathway does not appear to have been fully characterized: no HRMS and NMR spectra are in poor signal/noise ratio. The authors should consider taking advantage of the acidity of the N-acyl sulfomyl moiety, converting the final probe into its tertiary ammonium salt form to improve solubility and subsequently the quality of the NMR spectra.

---

## [Author Response]

Essential revisions:1) Please address if different ratios of the two enzymes expressed in one cell type be distinguished similarly to in different cells? This might be useful for gene expression studies, but would competition for substrates within a single cell cause problems?

An additional experiment has been carried out where FLuc_green and FLuc_red have been expressed at different levels in the same cell. Spectral bioluminescence imaging and unmixing has subsequently been performed to successfully reflect these differing expression levels with both LH_2_ and iLH_2_. Results of this experiment have been added to ‘Spectral unmixing of firefly luciferase mutants in vitro’, and experimental details to the Materials and methods section, Figure 2—figure supplement 2).

2) Can you better explain the spectral deconvolution, its limits, and address if it’s at all possible to separate the two signals based on a ratio or two single wavelength bands. If possible, what are the limitations of this for users who only have access to systems with band pass filters. In this context, it was felt that Figure 1 could be improved to incorporate such discussion and better clarify the design and goals of the study.

How spectral unmixing was performed is outlined in the Results section (‘Spectral unmixing of firefly luciferase mutants in vivo’), and further details can be found in the Materials and methods section (‘In vitro bioluminescence assays’, “In vivo bioluminescence imaging”). Due to the characterisation nature of this study, we always used 12-18 filters to ensure optimal spectral unmixing. However, we have performed further analysis of our in vitro data which has shown that even with a subset of these filters spectral unmixing of a similar accuracy can be achieved, meaning users only having access to systems with a limited set of bandpass filters will still be able to successfully use this dual-bioluminescence system (subsection “Spectral unmixing of firefly luciferase mutants in vitro”).

3) Please discuss how some of the best aspects of other published systems might be combined with the systems described here, for example, to do three colour BLI. Along these lines, in the second paragraph of the Discussion, is there a reason the click-beetle luciferase/naphthyl luciferin system wasn't used as benchmark? This system is mentioned in the Discussion and discounted for two-color work because of a broad emission peak. It does indeed have a broad peak, however, given the success of the spectral unmixing achieved in this study, we question whether the emission peak is so broad as to discount its use for multiplexing.

The potential use of this system for multi-coloured BLI has been discussed (Discussion, last two paragraphs), through the further mutagenesis of the Firefly luciferase protein to produce further colour mutants with iLH_2_ in the near infrared. Additional comments have been added to the Discussion section addressing other potential uses of this system combined with already published systems.

The click-beetle luciferase/naphthyl luciferin system was reported too late to be used as a comparison in this work. Without these substrates been included in this work we do not want to make any comparisons with regards to its spectral unmixing capabilities compared to iLH_2_ reported here. Additional comments have been added to the revised manuscript to clarify this (subsection “Spectral unmixing of firefly luciferase mutants in vitro”, Discussion).

4) The role of the structure in the study is not clear. The authors should better integrate the structural information or remove it to allow more of a focus on relevant information. Currently its stated that it may be valuable to improve the performance of infraluciferin, but was it of any direct use in the current work? This should be clarified.

Our multidisciplinary research team has been developing the near infrared dual colour imaging we present in this paper. We selected the journal *eLife* as it reports upon all aspects of the life sciences, to inform other researchers broadly in this field and at the periphery. We did not directly use this new structural information to derive improved novel luciferase mutants, as the focus of this work was to explore near-infrared bioluminescence imaging in vivo. In this particular work, the crystallisation and analysis of the luciferase/iLH_2_ complex has provided important insights into our understanding the phenomenon of the near infrared emission from iLH_2_. For example, the Arg337 H-bonding pattern differences in our 6HPS structure (compared to 4G36 containing the native luciferin inhibitor) appear to be necessary to accommodate the phenolic OH group of iLH_2_ in the same position as 4G36. It is interesting that a click beetle luciferase variant (CBR2), which contains an Arg to Ser mutation at equivalent residue 337, significantly red shifts the emission of an amino naphthyl analog that, like iLH_2_, is larger than native substrate luciferin ~ 65 nm.

In addition, other biological and physical scientists will find this informative and it will help others to design other near infrared bioluminescent emitters that will further bioluminescence imaging. We have added a sentence in the third paragraph of the Introduction and small addition to the Abstract to clarify this point.

5) The nature of the green and red mutants needs more description. It is assumed the green is the same as x11 + three additional mutations but it is not clear why this mutant was engineered. It is assumed that red is x11 + S284T + R354I (or is it E?) but it is not clear why this mutant was engineered. It is also a gap in the story that there isn't any biochemical analysis on these mutants (aside from spectra). Were they purified?

More description on the rationale for engineering luciferase mutants has been added to the revised manuscript (subsection “Spectral unmixing of firefly luciferase mutants in vitro”). The specific mutations contained in FLuc_green and FLuc_red are detailed in Materials and methods (subsection “Firefly luciferase mutants and cell lines”).As the mutations used have been previously reported (papers cited in manuscript), and the aim of this work was for use in vivo, all testing was carried out with the FLuc mutants expressed in the Raji B lymphoma cell line not with purified luciferases.

6) Its surprising that the very low photon yield from the infraluciferin system can enable collection of a useable signal. Can the authors comment on how the signal for infraluciferin is processed and if any special care was needed regarding substrate concentration/availability, camera settings, exposure time, binning or other strategies to enhance signal to noise. Along these lines, the authors used iLH_2_ concentration of 100mg/kg for their in vivo experiments. Could you go higher to enhance emission intensities? Is there toxicity data available, or would cost become a limitation?

The acquisition settings are stated in the Materials and methods section of the manuscript (subsection “In vivo model”). The only difference between imaging with LH_2_ and iLH_2_ is a longer acquisition time with iLH_2_ (a time of 120s which was determined by initial imaging using automatic acquisition mode on the Living Image software (Perkin Elmer)). We used a concentration of 100mg/kg of iLH_2_ for in vivo experiments to be consistent with the concentrations of LH_2_ used for imaging of in vivo models. No specific toxicity experiments were performed, but no adverse side effects were observed with iLH_2_ (subsections “In vitro bioluminescence assays” and “In vivo models”).

[Editors' note: further revisions were requested prior to acceptance, as described below.]

Reviewer #1:

The authors have addressed the major issues raised in the preview round of review as clearly described in the rebuttal letter. The change in the title confers to the manuscript a clear focus on a specific application. However, the manuscript still require some improvement since some points need to be better clarified.1) In the last paragraph of the subsection “Spectral characterisation of firefly luciferase mutants with LH_2_ and iLH_2_ in vivo” the authors comment that the relative intensities of the luciferases are comparable, thus the dynamic range of radiance values for both enzyme will be more similar giving a more accurate comparison of the process being monitored. The sentence results vague.

The sentence is a direct comparison of the relative intensities of Fluc_Green and Fluc_Red after administration of each substrate i.e. with LH_2_ Fluc_Red had a much higher photon count than Fluc_Green but with iLH_2_ the photon counts for Fluc_Red are similar to Fluc_Green and are therefore more comparable. I have added the wording in the subsection “Spectral unmixing of firefly luciferase mutants in vivo” to help make this clearer.

For example in the case one reporter is used under the control of a constitutive promoter and the second under a control of an inducible and much weaker promoter the fact that the luciferases have similar relative intensities would not help.

In our case both the Fluc_Green and Fluc_Red mutants have the same plasmid backbone and constitutive promotor, which is IRES. The cells have been also been sorted so that they are a 100% expressing population (see Materials and methods subsection “Firefly luciferase mutants and cell lines”). A study involving an inducible promotor would indeed lead to a reduction in expression compared to a constitutive promotor and therefore this would exhibit a similar presentation to LH_2_ i.e. one reporter having a much higher photon count than the other which will affect the accuracy of the spectral unmixing.

The authors proved that they could successfully separate in signals in cases where one reporter expression is around 10 times higher than the other (e.g. in the case of 90%/10% mixed population of cells). What happens if the expression is 99% and 1% for example? Will then the process be monitored with accuracy?

We chose known mixes of cell numbers that we were confident we could reliably count (in vitro) and inject (in vivo). The attenuation of green light is greater than red in biological tissue and therefore when Fluc_Green is shifted into the red by iLH_2_ this attenuation is reduced, resulting in a more accurate% when compared to the known cell numbers. However, light attenuation is an inherent problem of optical imaging and is dependent on depth, which is why bioluminescence can only be considered semi-quantitative. We have therefore validated our images by obtaining cells after imaging from the bone marrow and performing flow cytometry, which again confirms that iLH_2_ is more accurate than LH_2_.

2) In the proof of concept experiment using CAR T cells, the authors show that red signal is present in "green tumor only" animals, which is probably due to leakage of green signal in the red filters even after unmixing.

Mice are not imaged by a “green” and the “red” filter but through a sequence of 20 nm bandpass filter sets (for iLH_2_ we use 12 filters over 590-830nm) to obtain the spectrum which is used by the deconvolution algorithm to spectrally unmix. There is no leakage of green signal in the red filters.

This demonstrates that for this application accuracy in the separation of signals may be compromised. It would be helpful if the authors explain and comment in the Results and Discussion sections how they discriminate which red signal is representative of CAR T cells and which one is not. Is it maybe just necessary to image a "green tumor only" animal together with the CAR T cells treated one? This is important and need to be clarified for making readers able to reproduce the results.

Images 3 show the images obtained through each filter set and Image 4 shows the spectra obtained for Fluc_Green only when used to produce tumours in the systemic model. Figure 4 shows that Fluc_Green spectra when imaged as a single population provides distinctly different spectra than Fluc_Red. We therefore believe that the low level of “red” in the “green tumour only” mouse is most likely due to Fluc_Red photons from the treated mouse due its close proximity and sensitivity of the IVIS camera. I have therefore changed the wording of the sentence to “As would be expected, the tumour only control showed consistently high levels of FLuc_green signal which increased overtime compared to FLuc_red, which is likely due to the close proximity of Fluc_red photons emitted by the neighbouring treatment mouse.”

3) In the proof of concept experiments author shows an image at day 6 in which CAR T cells highly proliferate in tumors so red and green signal originate mostly in the same location in the animals. An image at earlier time point in which CAR T cells are in lymph nodes (for example 1-24 hours after injection) will allow the reader to see red and green signals in different locations in the animals. This is a minor point but will better illustrate the power of the dual color technology for cancer immunotherapy studies.

Although photons were detectable, images taken at day 1 after injection showed no Fluc_red signal. Whereas, images taken at day 3 showed Fluc_Red within the same regions as day 6 just at a lower level as indicated by the graph. This is due to the amount of CAR T cells initially injected and their expansion on CAR T cell activation. We injected 5x10^6^ CAR T cells which when diluted

throughout the body was not high enough to produce a photon image at day 1 due to attenuation. However, once the CAR T cells targeted the Raji tumours the CAR T cell activation causes local expansion and recruitment, thus giving signal only in these areas.

4) In the Materials and methods (subsection “in vivo bioluminescence imaging”) the authors report that imaging acquisition time is 120 sec. How long is the imaging session then using all the filters? More than 20 minutes? And is the emission of the reporters stable during this time? A comment about the length of the imaging session should be added in the Discussion.

Automatic acquisition was used for LH2 but 120 sec acquisition was used for iLH_2_ as the light produced is dimmer. 12 filters were used for iLH_2_ which takes 24 minutes. An open filter image was taken before and after to check that the total photon counts had not altered throughout this time. I have added the wording, “IVIS Spectrum (FOV 24, f/1, Medium (8)bin, automatic acquisition mode for imaging with LH_2_, FOV 24, f/1, Medium (8)bin, 120 s acquisition, total imaging time 24 mins for imaging with iLH_2_).” and “Open filter images were acquired prior to and post spectral imaging to confirm the stability of photon emission during spectral acquisition.”

5) In Figure 5 the percentage of red or green injected cells should be added to the unmixed images of the animals. The figure will result then clearer and of immediate interpretation.

The cell ratios are the same for all experiments and have therefore been represented by the green/red bars with the numbers on the corresponding graphs below. As the mice are not uniformly spaced adding the numbers to the bars gives the image a somewhat cluttered appearance and therefore even more confusing to the eye. For clarity we believe the diagram is better as is.

Reviewer #2:

The authors have done an acceptable (though not ideal) job of revising this manuscript. It is in my opinion suitable for publication as a marginally impactful, yet solid report.1) The manuscript needs editing from someone versed with the English language. For example, see incomplete sentence in Abstract (third sentence). As another example, though minor, the use of the word "origins" is awkward. The origin of luminescence emission is the photons released as a byproduct of the oxidative reaction between a luciferase and a luciferin substrate. The authors should consider an alternative word.

We thank the reviewer for their careful editorial eye.

- The sentence in the Abstract had been altered inadvertently and has been changed back to make a full sentence.

- We have changed the word ‘origin’ and have made a more accurate sentence ‘In this work, we explored the possible structural interactions in the enzyme that may account for the near infrared emission of iLH_2_ and its application to dual-BLI’.

- I am an English native, a senior academic who has a log track record in publishing accurate and high impact papers. As far as my poof reading skills go, I cannot see any other mistakes or ambiguities.

2) The report would benefit greatly if the authors were able to share more information about their substrate formulation, specifically for the in vivo experiments. What is the buffer? Are there any adverse effects from the substrate that the formulation was designed to address? What about solubility? Does it matter what you dissolve substrate in? In its current state I do not believe a reader could reproduce the in vivo experiments.

Stock solutions of LH_2_ or iLH_2_ where made up in sterile PBS. Sterile PBS is the recommended solute for LH_2_ for in vivo use although it can also be dissolved in sterile water. Using PBS we experience no solubility issues for iLH_2_ as it dissolved on gentle mixing by inversion the same as LH_2_. I have added the wording, “For imaging of in vivo models, LH_2_ and iLH_2_was solubilised in sterile PBS and animals were administered with substrate (2mg (or 100mg/kg) of either LH_2_ or iLH_2_ in 200μl or 400μl bolus respectively) via intraperitoneal (IP) injection.”

3) On another technical level, the final compound in the synthetic pathway does not appear to have been fully characterized: no HRMS and NMR spectra are in poor signal/noise ratio. The authors should consider taking advantage of the acidity of the N-acyl sulfomyl moiety, converting the final probe into its tertiary ammonium salt form to improve solubility and subsequently the quality of the NMR spectra.

The reviewer suggests a sensible method to try and get a more soluble compound. Unfortunately, we only have a few mg’s left and trying to form a salt on that scale would be virtually impossible. Repeating the synthesis is out of the question due to cost, time and restrictions on man power. I think the 1H NMR proves the identity of the product, especially bearing in mind the fully characterised starting material it is derived from. It is unfortunate we could not get a reasonable 13C NMR due to solubility issues. We also tried to obtain an accurate mass measurement under a variety of conditions and on different spectrometers, but did not obtain any M+ or sensible fragmentation that we could elucidate. This is unsatisfactory, however the precursor is fully characterised and there is no doubt of the compounds identity due to the 1H NMR and X-ray structure determination. I have added a sentence, “MS did not give M+ or meaningful fragment for accurate mass measurement”.